# Brazilian women's use of evidence-based practices in childbirth after participating in the Senses of Birth intervention: A mixed-methods study

Luísa da Matta Machado Fernandes[1]*, Sônia Lansky[2], Hozana Reis Passos[2], Christine T. Bozlak[3], Benjamin A. Shaw[3]

**1** Instituto René Rachou, Fundação Oswaldo Cruz (FIOCRUZ), Minas Gerais, Brazil, **2** Department of Health, Belo Horizonte, Minas Gerais, Brazil, **3** Department of Health Policy, Management, and Behavior, School of Public Health, University at Albany, State University of New York, Albany, NY, United States of America

* luisa@mattamachado.org

**Data Availability Statement:** We have reviewed the PLOS ONE requirement for data availability and

## Abstract

Brazil has a cesarean rate of 56% and low use of Intrapartum Evidence-based Practices (IEBP) of 3.4%, reflecting a medically centered and highly interventionist maternal health care model. The Senses of Birth (SoB) is a health education intervention created to promote normal birth, use of EBP, and reduce unnecessary c-sections. This study aimed to understand the use of intrapartum EBP by Brazilian women who participated in the SoB intervention. 555 women answered the questionnaire between 2015 and 2016. Bivariate analysis and ANOVA test were used to identify if social-demographic factors, childbirth information, and perceived knowledge were associated with the use of EBP. A qualitative analysis was performed to explore women's experiences. Research participants used the following EBP: birth plan (55.2%), companionship during childbirth (81.6%), midwife care (54.2%), freedom of mobility during labor (57.7%), choice of position during delivery (57.2%), and non-pharmacological pain relief methods (74.2%). Doula support was low (26.9%). Being a black woman was associated with not using a birth plan or having doula support. Women who gave birth in private hospitals were more likely not to use the EBP. Barriers to the use of EBP identified by women were an absence of individualized care, non-respect for their choices or provision of EBP by health care providers, inadequate structure and ambiance in hospitals to use EBP, and rigid protocols not centered on women's needs. The SoB intervention was identified as a potential facilitator. Women who used EBP described a sense of control over their bodies and perceived self-efficacy to advocate for their chosen practices. Women saw the strategies to overcome barriers as a path to become their childbirth protagonist. Health education is essential to increase the use of EBP; however, it should be implemented combined with changes in the maternal care system, promoting woman-centered and evidence-based models.

uploaded the minimal anonymized data set necessary to replicate our quantitative study findings to the Dryad public repository (https://datadryad.org/stash); the current DOI of the data set is doi:10.5061/dryad.r7sqv9sb8. The data available correspond to the quantitative data used, however the qualitative data used can not be made available to the public, considering the sensitivity of the data. The majority of the women answering the open-ended questions that composed the qualitative data used their proper names and last names of family members, physicians, midwifes and doulas. Therefore, following the requirements of our IRB approval and our ethical commitment with the women who answered the survey, we cannot make the qualitative data publicly available without compromising the women´s and health professionals' identities. In this case, de-identifying the data by taking the names out of the answers would compromise the answers and quality of the data analysis. Therefore, the qualitative data will be available upon request to the Senses of Birth Steering Committee who will ensure the open-ended questions are used according to the IRB requirements. The request to the steering committee can be directed to the co-principal investigators: Dr. Bernardo Oliveira, at the School of Education from the Federal University of Minas Gerais. Contact information: posfaebernardooliveira@gmail.com; and Dr. Sônia Lansky, at the Belo Horizonte´s Department of Health. Contact information: sonialansky@gmail.com

**Funding:** This work was supported by funding from a group of organizations list bellow, part of the call MCTI/CNPq/MS/SCTIE/Decit/Fundação Bill e Melinda Gates N o 05/2013 Organizations: - National Council of Technological and Scientific Development (CNPq)- http://www.cnpq.br/web/guest/fomento-tecnologico, - Ministry of Health - https://www.saude.gov.br/trabalho-educacao-e-qualificacao/gestao-e-regulacao-do-trabalho-em-saude/premio-inovasus, - Bill and Melinda Gates Foundation -https://www.gatesfoundation.org/, - Pan-American Health Organization - https://www.paho.org/bra/, - Funding Agency of the State of Minas Gerais (FAPEMIG) - https://fapemig.br/pt/. The funding was directed to implement the health education intervention and data collection. Authors did not receive personal funding. The funders had no role in study design, data collection and analysis, decision to publish, or preparation of the manuscript.

**Competing interests:** The authors have declared that no competing interests exist.

## Introduction

The use of evidence-based maternity care has been recommended by the Pan-American Health Organization (PAHO) and World Health Organization (WHO) since 1985 and 1996, respectively, and reinforced by the guidelines in 2018 [1–4]. Evidence-based practices (EBP) are derived from the use of the best available research results to guide health care practices [2]. Increasing the use of EBP during labor and childbirth can optimize maternal, fetal, and new-born outcomes, support effective and respectful care, and assist providers' and women's deci-sions [2, 5, 6]. Access to evidence-based care for all women is considered a fundamental reproductive right, based on the notion that quality maternal care and delivery should be humane and dignified [5].

A positive childbirth experience includes the use of EBP that promotes a safe environment for labor and delivery, with support from trained health professionals, resulting in positive clinical outcomes for mother and baby [3]. No less relevant, the experience should fulfill a woman's prior personal and socio-cultural beliefs and expectations, promoting a sense of per-sonal achievement and control through childbirth, allowing for participation in all decision-making processes, with true informed consent [3]. Positive childbirth experiences are consis-tent with at least two of the Sustainable Development Goals (SDG): Goal #3 intends to ensure healthy lives and promote well-being for all ages, and Goal #5 addresses gender equality focused on ensuring universal access to sexual and reproductive health and reproductive rights [7].

The overwhelming majority of studies identifying barriers to use EBP have focused on the health professional's perspective and/or medical records data [8–12], with only a few studies including the women's perspectives [13–15]. Identified barriers to using the practices were lack of a set of robust maternity performance measures with a buy-in of key stakeholders for measuring, perverse incentives of payment systems, limited reliance on best evidence in lead-ing guidelines for maternity care, insufficient knowledge among women and health care pro-fessionals, limitations of views put forth in media and popular discourse, adverse effects of pressure from the health care industry, health professionals not offering the methods, and poor hospital infrastructure [8–15].

In Brazil, nationwide studies about the use of maternal care EBP are limited. The Birth in Brazil Survey, a nationwide hospital-based study that interviewed over 23,000 mothers and reviewed hospital records on live births between 2011 and 2012, found that only 3.4% of live births utilized best practices recommended by the WHO during labor and childbirth [16, 17]. Meanwhile, non-recommended practices were frequently used, such as Kristeller (a maneuver used by professionals to pull the baby), identified in 36.5% of the vaginal deliveries [16].

Brazil's high cesarean rate offers a glimpse into its medically-centered and highly inter-ventionist maternal health care model [18, 19]. Brazil has a universal public health system named the Unified Health System and known as "SUS". SUS is a tax-funded health care system based on government-owned services while also relying upon nonprofit and for-profit pro-vider contractors when needed. SUS offers universal coverage, comprehensive care, and equity of access to health services within a regionalized hierarchical and decentralized system without any additional costs to the population [20–22]. Concomitantly, a national agency regulates the private health system funded by employers or individuals who buy private insurance and/or use out-of-pocket payments for services under a specific set of requirements and a defined pro-vider network [20, 23]. Currently, 74% of the population depends exclusively on SUS to access health care, and 26% are enrolled in private health insurance [20]. However, 58% of births in Brazil occur at private hospitals, with a considerably higher average of cesarean rates than the national average (83% vs. 56%) [24, 25]. Moreover, most women have access to health facilities

and skilled health professionals, with 99.4% of pregnant women having at least one prenatal care consult and an estimated rate of 99% of births attended by skilled health professionals [26].

Despite the high rate of women accessing maternal care, regional differences impacting access and quality of care are a reality in Brazil [5, 27]. When those are combined with social inequalities, they contribute to situations where there is a lack of access to appropriate and timely care, referred to as a "too little too late" (TLTL) scenario [5, 27]. In contrast to the TLTL, the "too much too soon" (TMTS) scenario of maternal care intervention, also potentially dangerous, is often encountered in Brazil, promoting a childbirth paradox of care [5, 27, 28]. TMTS is defined as unnecessary early maternal health interventions that reduce the spontaneous birth likelihood [5, 27]. The TMTS scenario is often a symptom of healthcare policies that approach labor and childbirth as a medical event and not as a physiologic process with its own social and cultural aspects [5].

Brazil's current maternal national health policy, named *"Rede Cegonha"*, was instituted in 2011 as a strategy to implement a network of care that supports women's reproductive rights and promotes humanized care [29]. The *"Rede Cegonha"* reasserts other maternal and childbirth health policies previously implemented, promoting multidisciplinary teams, updated protocols, and monitoring health indicators with target-coupled funding, while also incorporating principles of the Childbirth Humanization Movement in Brazil [1, 16, 30]. The Childbirth Humanization Movement is the Brazilian arm of the Respectful Maternity Care (RMC) international movement, which advocates for respectful care as a universal human right for every childbearing woman in every health system around the world [31].

Thus, considering the need to increase the use of EBP and improve the childbirth experience, this study aimed to describe the Brazilian women experience related to the use of intrapartum EBP after participating in the health education intervention, the Senses of Birth (SoB) intervention.

## Materials and methods

The study is part of the research project named "Senses of Birth: Effects of the interactive intervention on the perception changes on labor and childbirth" [32] and used a mixed-methods research design to analyze associations between social-demographic characteristics, childbirth information, perceived knowledge regarding the EBP, Normal Birth and Cesarean and women's behavior. The current study was approved by The Federal University at Minas Gerais Institutional Review Board (IRB)—COEP/UFMG (934.472) and by the University at Albany IRB (18-X-209-01). All women provided written informed consent prior to answering the post-intervention survey and the follow-up survey.

### The SoB health education intervention

The SoB intervention was created to address the overmedicalization of maternal care and excessive cesarean rates in Brazil [32], consistent with the Childbirth Humanization and RMC movements principles [1, 31]. The intervention, set in public spaces such as parks and street fairs, uses holograms, videos, role-playing, and other interactive techniques to offer the participants a childbirth experience. All participants are invited to walk through different sets as if pregnant and face the medicalized industries' offers for childbirth. A detailed description of the intervention settings, methods, and implementation can be found in previously published papers of the Senses of Birth research team [32, 33].

Among the maternal care topics approached in the intervention are the intrapartum evidence-based practices. Evidence supporting these intrapartum practices is summarized in the

WHO intrapartum care recommendations for a positive childbirth [3]. These practices are primarily, although not exclusively, dependent on the patient's engagement. Thus, the SoB intervention was designed to address practices over which pregnant women have some degree of control or choice.

The intrapartum EBP discussed in the SoB intervention, and part of this study are: 1) creation and use of a birth plan; 2) one-to-one continuous support, including doula support and/or a companionship of choice throughout childbirth; 3) midwife care; 4) use of non-pharmacological pain relief methods; 5) freedom of mobility throughout labor; and 6) choice of position at delivery.

SoB is grounded in the Theory of Planned Behavior (TPB), and its proposed framework can be found in Sup. 1. TPB states that behavior can be directly influenced by the intention to engage in that behavior [34, 35]. The intention to perform a behavior is composed of attitudes, subjective norms, and perceived control over the behavior [35, 36]. Others have used TPB to understand childbirth and preference for the type of birth [37–40], although no studies were found focusing on its use with intrapartum EBP.

Perceived behavioral control is the individual's perceived self-efficacy, that is, the confidence in their ability to perform the behavior [35, 36]. Such perception of control is built by internal factors (knowledge acquired and skills learned) and external factors (practical resources available, opportunities to use it, and the presence of other supportive conditions) [34–36]. For a woman to perceive that she can control the behavior of using the intrapartum EBP, therefore, increasing her intention to use it, she will need knowledge, skills, resources, and opportunities to increase her self-efficacy. The internal and external factors can facilitate or create challenges for a woman to use an EBP. As observed by previous results of the SoB impact on women's knowledge, Brazilian women have moderate knowledge about normal birth and cesarean, and weak knowledge about EBP, although most perceived an increase in all three knowledge domains (normal birth, cesarean, and evidence-based practices) after the SoB intervention [41]. The impact of the SoB intervention on women's perceived knowledge indicates a need to improve information regarding normal birth and cesarean, and an even higher need to discuss the use of evidence-based practices with pregnant women [41].

## Data and sample

Eligible participants were identified at the Senses of Birth Intervention entrance when providing an affirmative response to the current pregnancy question. They were invited to join the study by one of the trained interviewers. All pregnant women over 18 years old who visited the intervention between March 2015 and March 2016 were invited to answer the post-intervention survey, forming a convenience sample of 1,287 women [32]. Only six women refused to answer it. The group of pregnant women who responded to the post-intervention survey were also invited to join the online follow-up survey. Five hundred and fifty-five women answered the follow-up post-partum survey between June 2015 and April 2016, with a response rate of 43.1%, in three different states and five different cities of Brazil (Belo Horizonte/MG; Rio de Janeiro/RJ and Niterói/RJ; Ceilândia/DF and Brasília/DF) [32]. Forty-four percent of those who answered the follow-up survey were included in the qualitative analysis, for a total sample of 258 (Fig 1). On average, women answered the follow-up survey three months after giving birth (57.4%), ranging from 0 to 29 months, with 95.2% of women responding within seven months of childbirth.

Only women over 18 years old who answered the post-intervention survey and follow-up survey were included in the present study. After three emails and three phone call attempts, women who did not respond to the follow-up survey were excluded. For the sub-set of women

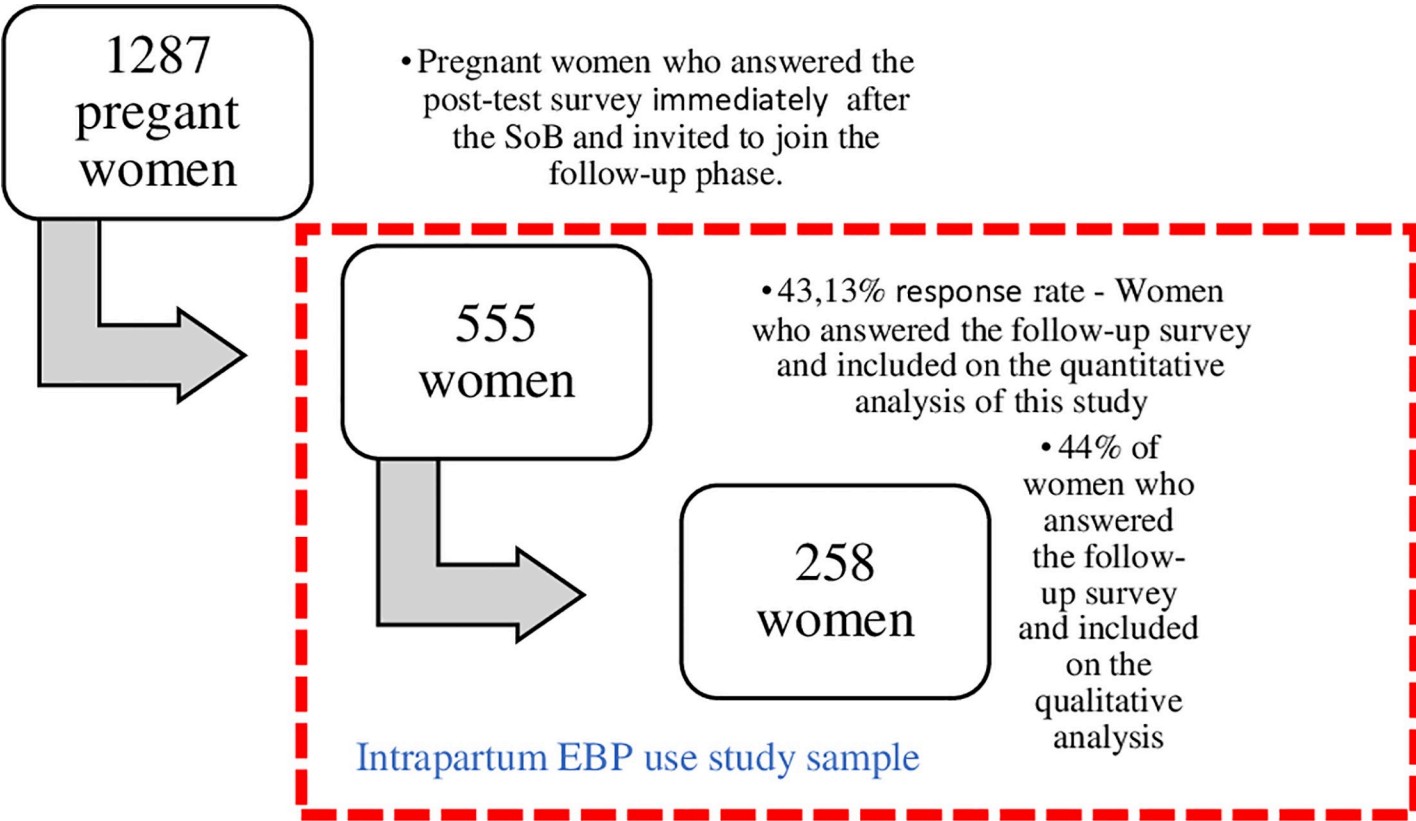

**Fig 1. Data collection timeline and sample of women who joined the Senses of Birth intervention and answered the follow-up survey.**

included for the qualitative portion of the study, two exclusion criteria were applied: a) Answering two or fewer sentences to the question "Tell us a little about your birth experience;" b) Blank answers or "no comments" statements without considerable information regarding the childbirth to the subsequent open-ended questions.

## Quantitative measures

The follow-up post-partum survey was an online self-administered structured questionnaire used to collect quantitative and qualitative data (S2 and S3 Files). The post-partum instrument contained questions about labor and childbirth experience, use of EBP during childbirth, obstetric violence, breastfeeding, and memory of the educational intervention. The survey was linked to the post-intervention instrument to assess socioeconomic, demographic, and perceived knowledge information. Four groups of variables were selected for the quantitative analysis in this study:

1. **Socio-demographic characteristics.** Age, race, education, private health insurance, and income.

2. **Obstetric characteristics.** First pregnancy, type of hospital, type of birth, perceived ability to have a normal birth.

3. **Perceived Knowledge.** The perceived knowledge variables were grouped into three different domains, based on the results of factor analysis of a previous study [41]. The domains are: 1) EBP Knowledge, 2) Normal Birth Knowledge, and 3) Cesarean Knowledge. For each knowledge domain, a mean score before the intervention (based on recall) and a mean

score after the intervention were computed from the sum of all variables in that domain, ranging from 1 to 5 points [41]. A specific change score was calculated for each domain, representing the women's perceived variation of knowledge using the difference between the mean after and the mean before, ranging from -5 to 5 [41]. Women who did not perceive an increase in knowledge after the intervention ranked within -5 and 0. Women who perceived knowledge increase after the intervention ranked between 0.1 and 5 [41].

4. **Use of Intrapartum EBP.** Participants were asked whether or not they used each of the evidence-based practices during labor and delivery: birth plan; companionship during childbirth; doula support; midwife care; freedom of mobility during labor; choice of position during delivery; use of non-pharmacological methods for pain relief.

Further information describing the variables used, local context, and variable preparation can be found in S4 File.

## Quantitative analysis

To identify how social-demographic factors, childbirth information, and perceived knowledge were associated with the use of evidence-based practices during labor and childbirth, chi-square, and ANOVA tests were performed. Associations were considered statistically significant with p-values ≤ 0.05. All variables presented normal distributions. The statistical program IBM SPSS Statistics 24R was used for data analysis.

## Qualitative measures

The qualitative data were collected through seven open-ended follow-up survey questions listed below (Fig 2). All questions and answers were in Portuguese, the women's native language. Women's responses were translated and analyzed in English by bilingual researchers; any necessary adjustments for clarity were made. However, cultural references were kept in Portuguese and explained in the sequence. The first question was required to proceed with the survey; the subsequent ones were optional. The open-ended questions unfolded from the closed-ended questions, allowing women to express their opinions, feelings, and perspectives. The set of open-ended written responses of each woman was the unit of analysis.

## Qualitative analysis

To explore women's experiences regarding the use of intrapartum EBP, identifying barriers, facilitators, and strategies, a qualitative analysis of the open-ended questions was performed.

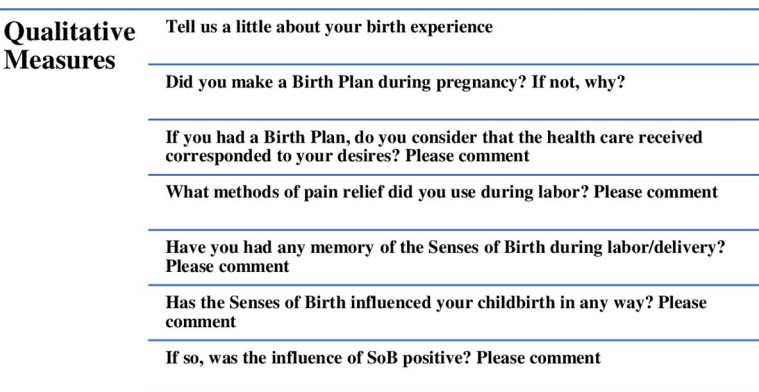

**Fig 2. Open-ended questions answered by women on the follow-up SoB survey.**

An inductive open-coding analytic process was conducted, with motivated line-by-line reading without previously established categories allowing the researcher to identify events that could become the basis of categorization. A second researcher reviewed all the codes and quotations to validate the codes and make sure coding was consistent through all interviews. Themes and categories emerged from the codes with an agreement between the two researchers. Disagreements were discussed until a consensus was reached. The software used for the qualitative analysis was Atlas.ti version 8.3.1.

Each EBP described by the women was considered a theme, and any reference to those was coded: birth plan, doula support, midwife care, companionship during childbirth, freedom of mobility during labor, choice of position during delivery, and/or use of non-pharmacological methods for pain relief. Forty-five characterizing codes emerged from the open-coding analysis and were grouped into three categories: Outcomes, Barriers, Facilitators/Strategies. Outcomes refer to the group of codes that describe the woman's perspective of using or not using one or more EBP. Barriers refer to the group of codes that describe barriers a woman identified to using or not using at least one EBP. Facilitators/Strategies refer to the group of codes that describe what elements the women identified as support for the use of EBP. Discourse Analysis was used to support triangulation of analysis, advancing the understanding of women's experience with intrapartum EBP.

## Results

The qualitative and quantitative results are presented in an integrated manner. The categories that emerged from the qualitative analysis guide the display of results: outcomes, barriers, and facilitators/strategies to use the intrapartum EBP.

Women who answered the follow-up survey (555) were predominantly young (< 34 years old; 82.7%), black (53.2%), with more than 13 years of education (76.3%), and had private health insurance (78.8%) (Table 1). Approximately a third (32.6%) had a family monthly income between 2 to < 5 MW. The majority of the women had a vaginal birth (54.1%), gave birth at a private hospital (63.9%), had one or more previous pregnancies (52.1%), and believed they were capable of having a normal birth (93.6%) (Table 1). Women also perceived an increase in their knowledge after participating in the intervention for all domains: EBP Knowledge (85.4%), Normal Birth Knowledge (65.8%), and Cesarean Knowledge (64.2%) (Table 1).

Previous findings have identified that the 555 women who answered the follow-up survey had similar socio-demographic characteristics as the 1,287 women who answered the post-intervention survey, with the only significant difference in the increased perceived knowledge before and after the intervention for the three domains [42]. Before and after the intervention, the perceived knowledge was higher among women who answered the follow-up survey compared to all women who answered the post-intervention survey [42]. The characteristics of the 258 women included in the qualitative analysis were similar to the total sample (n = 555), except for education: 82.0% of the women within the qualitative analysis had 13 years or more of formal education, compared to 76.3% of the total sample (Table 1).

### Women's perceived outcomes of using intrapartum EBP

The majority of women used the EBP presented in the intervention: birth plan (55.2%), companionship during childbirth (81.6%), midwife care (54.2%), freedom of mobility during labor (57.7%), choice of position during delivery (57.2%), and non-pharmacological pain relief methods (74.2%) (Table 2). Doula support was used by 26.9% of women (Table 2). Women who were included in the qualitative analysis had even higher use of EBP. The women who had midwife care, freedom of mobility during labor, choice of position at delivery, and used

**Table 1.  Characteristics of women who participated in the SoB intervention (follow-up and qualitative analysis).**  Brazil 2015–2016.

| Characteristics | Follow-up Survey | Qualitative sample | |
|---|---|---|---|
| | 555 Total[1] N (%) | 258 Total[1] N (%) | P value |
| **Age** | | | |
| 19–34 years | 455 (82.7) | 214 (83.9) | 0.45 |
| ≥ 35 years | 95 (17.3) | 41 (16.1) | |
| TOTAL | 550 | 255 | |
| **Education** | | | |
| < 12 years | 131 (23.7) | 46 (18.0) | 0.006* |
| ≥ 12 years | 421 (76.3) | 209 (82.0) | |
| TOTAL | 552 | 255 | |
| **Income[2]** | | | |
| < 2 MW | 102 (19.7) | 36 (15.3) | 0.271 |
| 2 to < 5 MW | 169 (32.6) | 80 (34.0) | |
| 5 to < 10 MW | 135 (26.0) | 63 (26.8) | |
| ≥10 MW | 113 (21.8) | 56 (23.8) | |
| TOTAL | 519 | 235 | |
| **Race** | | | |
| White | 257 (46.6) | 121 (47.5) | 0.839 |
| Black and Others | 295 (53.2) | 134 (52.5) | |
| TOTAL | 552 | 255 | |
| **Private Health Insurance** | | | |
| Yes | 436 (78.8) | 209 (81.6) | 0.250 |
| No | 117 (21.2) | 47 (18.4) | |
| TOTAL | 553 | 256 | |
| **Type of Hospital** | | | |
| SUS[3] (Public) | 200 (36.1) | 85 (32.2) | 0.325 |
| Private | 354 (63.9) | 171 (67.8) | |
| TOTAL | 554 | 256 | |
| **Type of Birth** | | | |
| Vaginal | 300 (54.1) | 149 (58.2) | 0.063^ |
| Cesarean | 255 (45.9) | 107 (41.8) | |
| TOTAL | 555 | 256 | |
| **First Pregnancy** | | | |
| Yes | 234 (47.9) | 119 (52.7) | 0.076^ |
| No | 255 (52.1) | 107 (47.3) | |
| TOTAL | 489 | 226 | |
| **Able to have normal birth** | | | |
| Yes | 516 (93.6) | 242 (94.9) | 0.183 |
| No | 35 (6.4) | 13 (5.1) | |
| TOTAL | 551 | 255 | |
| **Normal Birth Knowledge Domain** | | | |
| Perceived Knowledge Increased | 366 (65.9) | 163 (63.7) | 0.467 |
| Perceived Knowledge Not Increased | 189 (34.1) | 93 (36.3) | |
| TOTAL | 555 | 256 | |
| **Cesarean Knowledge Domain** | | | |
| Perceived Knowledge Increased | 436 (78.8) | 162 (63.8) | 0.998 |
| Perceived Knowledge Not Increased | 197 (35.8) | 92 (36.2) | |
| TOTAL | 550 | 254 | |

*(Continued)*

**Table 1.** (Continued)

| Characteristics | Follow-up Survey | Qualitative sample | |
|---|---|---|---|
| | 555 Total[1] N (%) | 258 Total[1] N (%) | P value |
| **EBP Knowledge Domain** | | | |
| Perceived Knowledge Increased | 455 (85.4) | 203 (83.2) | 0.271 |
| Perceived Knowledge Not Increased | 78 (14.6) | 41 (16.8) | |
| TOTAL | 533 | 244 | |

[1]Total varies dues to missing data for each variable.

[2]Monthly Minimum Wage in 2015: R$788.00 = U$224.14.

[3]SUS–Unified Health System.

Chi-Square: ^P-value ≤ 0.1 *P-value ≤ 0.05.

**Table 2. Use of EBP among women who participated in the SoB intervention (follow-up and qualitative analysis).** Brazil 2015–2016.

| Evidence-Based Practices | Follow-up Survey | Qualitative sample | |
|---|---|---|---|
| | 555 Total[1] N (%) | 258 Total[1] N (%) | P value |
| **Birth Plan** | | | |
| Yes | 306 (55.2) | 143 (56.1) | 0.773 |
| No | 248 (44.8) | 112 (43.9) | |
| Total | 553 | 255 | |
| **Companionship during Childbirth** | | | |
| Yes | 453 (81.6) | 206 (83.4) | 0.510 |
| No | 83 (15.5) | 41 (16.6) | |
| Total | 536 | 247 | |
| **Doula Support** | | | |
| Yes | 146 (26.9) | 74 (29.7) | 0.183 |
| No | 396 (73.1) | 175 (70.3) | |
| Total | 542 | 249 | |
| **Midwife Care** | | | |
| Yes | 294 (54.2) | 157 (63.1) | 0.000** |
| No | 248 (45.8) | 92 (36.9) | |
| Total | 542 | 249 | |
| **Freedom of Mobility during labor** | | | |
| Yes | 248 (57.7) | 139 (71.2) | 0.000** |
| No | 182 (42.3) | 56 (28.7) | |
| Total | 430 | 195 | |
| **Choice of Position at delivery** | | | |
| Yes | 246 (57.2) | 129 (66.2) | 0.000** |
| No | 184 (42.8) | 66 (33.8) | |
| Total | 430 | 195 | |
| **Non-Pharmacological Pain Relief Methods** | | | |
| Yes | 316 (74.2) | 160 (82.5) | 0.000** |
| No | 110 (25.8) | 34 (17.5) | |
| Total | 426 | 194 | |

[1]Total varies dues to missing data for each variable.

Chi-Square

**P-value ≤ 0.001.

non-pharmacological pain relief methods were more likely to answer the open-ended questions, describing their childbirth experience. Furthermore, all women included in the qualitative analysis referred to at least one evidence-based practice in their answers to the open-ended questions.

Women described positive outcomes after using the intrapartum EBP, primarily by those who had a vaginal birth. In the bivariate analyses, vaginal birth was associated with the use of all the EBP studied: birth plan (58.8%, $p \leq 0.05$); the companionship of choice during childbirth (56.7%, $p \leq 0.05$), doula support (80.8% $p \leq 0.01$); midwife care (64.6%, $p \leq 0.01$); freedom of mobility during labor (69.4%, $p \leq 0.01$); choice of position during delivery (68.7%, $p \leq 0.05$); and use of non-pharmacological methods for pain relief (70.9%, $p \leq 0.01$) (Table 3). Two participants who had a vaginal birth in different hospitals described the outcomes after or related to using different EBPs. The woman who gave birth in a private hospital described, *"I had a natural birth in the water, really calming. My husband was accompanying me, and my doula, so it was very comforting and pleasurable"*. In contrast, a participant who gave birth in a public hospital reported, *"My desire for childbirth as natural as possible was respected, even considering the duration of the labor. Because I was a first-time mother, I went to the hospital very early, in the early stages of pain. Also, the support provided by the hospital structure and its staff went far beyond our expectations."*

Among the 555 women who participated in this study, 27.2% had a scheduled cesarean before labor onset, with limited possibility to use the described EBPs. Among the 253 women who had a cesarean, 40.3% had an intrapartum cesarean, initiated labor before the surgery, and could have had the opportunity to use the practices during labor. Women who had a cesarean and perceived it as an autonomous decision or a needed intervention, described the use of EBP, such as the companionship of their choice, with a positive connotation: *"I felt calm during childbirth because the doctor was reliable, and she accompanied me during pre-natal care. My husband was by my side the whole time. A cesarean section was performed after a decision I made with my husband and my doctor."*

Notably, using one or more EBP was associated with a positive childbirth experience, which women described as feeling satisfied with the birth and as having a comforting/calming experience during childbirth: *"We were walking down the hallway to help the labor progress. It felt so good to see the people there and to overcome the pain that came. I felt like a warrior."* Women who had a vaginal birth expressed feeling safe with the presence of midwives and doulas. Women have also reported that their expectations regarding childbirth were attended to when using a birth plan, as exemplified by the following quotation: *"Since I had the doula by my side the whole time, I felt very safe! I had a massage, used the ball and the shower, played background music, and had freedom of movement."*

The one-to-one continuous support practices (doula and companionship of choice) were considered by women who had a vaginal birth as practices that helped them demystify normal birth, enrich childbirth experience, and overcome fears. It is noticeable that the majority of women described the use of EBP as a bundle of practices, especially non-pharmacological pain relief methods, freedom of mobility during labor, and choice of position during delivery. One woman described how having doula support resulted in using other EBP: *"It felt free, exactly how I wanted. The cool thing was the doulas also suggested things that I did not think would help me, and it did, like using the birth ball."*

The use of different EBP gave women the perception of self-efficacy, strength, and courage. The experience is described as gaining control over their bodies. They also described having their desires respected and expectations attended to when the practices were used, independently of the type of birth. Two women who had vaginal births exemplify those findings describing their feelings of control over their bodies, their choices respected, and their

**Table 3. Use of intrapartum evidence-based practices among women in the Senses of Birth intervention who participated in the follow-up survey, by type of hospital, type of birth, and parity.** Brazil, 2015–2016.

| | Type of Hospital Total N = 554[2] | | | Type of Birth Total N = 555[2] | | | First Pregnancy Total N = 489[2] | | | Able to have a Normal Birth Total N = 551[2] | | |
|---|---|---|---|---|---|---|---|---|---|---|---|---|
| | Public Hospital[1] N (%) | Private Hospital N (%) | P value | Vaginal N (%) | Cesarean N (%) | P value | Yes N (%) | No N (%) | P value | Yes N (%) | No N (%) | P value |
| **Birth Plan** | | | | | | | | | | | | |
| Yes | 115 (37.5) | 190 (62.3) | 0.404 | 180 (58.8) | 126 (41.2) | 0.014* | 135 (48.9) | 141 (51.1) | 0.627 | 291 (95.4) | 14 (4.6) | 0.057^ |
| No | 85 (34.3) | 163 (65.7) | | 120 (48.4) | 128 (51.6) | | 99 (46.7) | 113 (53.3) | | 224 (91.4) | 21 (8.6) | |
| Tot. | 200 (36.2) | 353 (63.8) | | 300 (54.2) | 254 (45.8) | | 234 (47.9) | 255 (52.1) | | 516 (93.6) | 35 (6.4) | |
| **Companionship during Childbirth** | | | | | | | | | | | | |
| Yes | 168 (37.2) | 284 (62.8) | 0.038* | 257 (56.7) | 196 (43.3) | 0.000** | 194 (48.0) | 210 (52.0) | 0.932 | 425 (94.2) | 26 (5.6) | 0.338 |
| No | 21 (25.3) | 62 (74.7) | | 27 (32.5) | 56 (67.5) | | 34 (48.6) | 36 (51.4) | | 75 (91.5) | 7 (8.5) | |
| Tot. | 189 (35.3) | 346 (64.7) | | 284 (53.0) | 252 (47.0) | | 228 (48.1) | 246 (51.9) | | 500 (93.8) | 33 (6.2) | |
| **Doula Support** | | | | | | | | | | | | |
| Yes | 76 (52.4) | 69 (47.6) | 0.000** | 118 (80.8) | 28 (19.2) | 0.000** | 58 (45.3) | 70 (54.7) | 0.364 | 141 (97.9) | 3 (2.1) | 0.022* |
| No | 119 (30.1) | 277 (69.9) | | 178 (44.9) | 218 (55.1) | | 175 (50.0) | 175 (50.0) | | 365 (92.6) | 29 (7.4) | |
| Tot. | 195 (36.0) | 346 (64.0) | | 296 (54.6) | 246 (45.4) | | 233 (48.7) | 245 (51.3) | | 506 (94.2) | 31 (5.7) | |
| **Midwife Care** | | | | | | | | | | | | |
| Yes | 146 (49.8) | 147 (50.2) | 0.000** | 190 (64.6) | 104 (35.4) | 0.000** | 117 (44.8) | 144 (55.2) | 0.060 | 281 (95.9) | 12 (4.1) | 0.047* |
| No | 49 (19.8) | 199 (80.2) | | 106 (42.7) | 142 (57.3) | | 116 (53.5) | 101 (46.5) | | 225 (91.8) | 20 (8.2) | |
| Tot. | 195 (36.0) | 346 (64.0) | | 296 (54.6) | 246 (45.4) | | 233 (48.7) | 245 (51.3) | | 536 (93.7) | 36 (6.2) | |
| **Freedom of Mobility during labor** | | | | | | | | | | | | |
| Yes | 120 (48.6) | 127 (51.4) | 0.000** | 172 (69.4) | 76 (30.6) | 0.013* | 107 (47.3) | 119 (52.7) | 0.939 | 236 (95.5) | 11 (4.5) | 0.231 |
| No | 47 (25.8) | 135 (74.2) | | 105 (57.7) | 77 (42.3) | | 81 (52.3) | 74 (47.7) | | 169 (92.9) | 13 (7.1) | |
| Tot. | 167 (38.9) | 262 (61.1) | | 277 (64.4) | 153 (35.6) | | 188 (49.3) | 193 (50.7) | | 405 (94.4) | 24 (5.6) | |
| **Choice of Position at delivery** | | | | | | | | | | | | |
| Yes | 115 (46.9) | 130 (53.1) | 0.000** | 169 (68.7) | 77 (31.3) | 0.032* | 103 (46.0) | 78 (49.7) | 0.477 | 234 (95.5) | 11 (4.5) | 0.251 |
| No | 52 (28.3) | 132 (71.7) | | 108 (58.7) | 76 (41.3) | | 121 (54.0) | 79 (50.3) | | 171 (92.9) | 13 (7.1) | |
| Tot. | 167 (38.9) | 262 (61.1) | | 277 (64.4) | 153 (35.6) | | 224 (58.8) | 157 (41.2) | | 405 (94.4) | 24 (5.6) | |
| **Non-Pharmacological Pain Relief Methods** | | | | | | | | | | | | |
| Yes | 154 (48.9) | 161 (51.1) | 0.000** | 224 (70.9) | 92 (29.1) | 0.000** | 135 (47.2) | 151 (52.8) | 0.848 | 300 (95.2) | 15 (4.8) | 0.181 |
| No | 11 (10.0) | 99 (90.0) | | 50 (45.5) | 60 (54.5) | | 44 (48.4) | 47 (51.6) | | 101 (91.8) | 9 (8.2) | |
| Tot. | 165 (38.8) | 260 (61.2) | | 274 (64.3) | 152 (35.7) | | 178 (47.3) | 198 (52.7) | | 401 (94.4) | 24 (5.6) | |

[1]Public Hospital is the SUS (Unified Health System) Hospitals.

[2]Total varies due to missing data for each variable.

Chi-Square

^P-value ≤ 0.1

*P-value ≤ 0.05

**P-value ≤ 0.001.

perceived self-efficacy. **1)** "*I let my body guide the birth; there was nothing that would inter-fere with it. I ate, walked, lay down, did everything I wanted during labor. I felt safe, calm, and I loved the feeling and observing my body working*"; 2) "*They left me free to do whatever I wanted; it was my own labor. It was a wonderful moment.*"

Women who had a cesarean or a vaginal birth without access to an EBP described their experience with dissatisfaction, as indicated by one who had a vaginal birth at a private hospi-tal: "*I wanted a vertical position, but in the expulsive, I ended up in lithotomy.*" Not having the companionship of choice at all times was associated with increased anxiety, loneliness, fear, frustration, and dissatisfaction, as a woman who had a cesarean describes: "*I was taken to the surgical unit, anesthetized, and left unaccompanied for almost two hours until delivery. I did not like this period, being in a cold room, alone and anxious for my first childbirth.*"

Not having a birth plan or having the birth plan disrespected was described with regret by women who had a cesarean, such as this one who reported increased knowledge after the SoB intervention: *When I learned [about the birth plan] I was already living a moment of so many changes that I could not do it, I regret not having done it, but I really did not know before the intervention.*" Frequently, women who did not use a birth plan or non-pharmaco-logical pain relief methods also described not having a choice of position at delivery, no doula support, or companionship of choice at all times.

## Perceived barriers to using intrapartum EBP

Women regularly described the hospital conduct, protocols, and the EBP not being offered as reasons for not using one or more EBP, as one of them explicitly stated: *"They made me feel comfortable about choosing what I wanted to do, but they did not offer me those options. I stayed quiet on the bed."*

Furthermore, women described they were not offered the non-pharmacological methods for pain relief as the reason for not using this practice—*"Nothing was offered at the hospital, only anesthesia."* The situation occurred in public and private hospitals, indicating that women would have used it if the hospital had offered the methods as a regular practice. Birth plan, choice of position at delivery, and freedom of mobility throughout labor were also EBP not used for lack of choice or due to not being offered. One of the women described how she felt trapped and silenced by the hospital protocol that did not offer her EBP: *"Because I was not given the option. When you arrive and are admitted to a maternity ward, there is an implicit agreement that you will follow the on-call attendant choices. The pregnant woman is not heard concerning her expectations or wishes."*

Nonetheless, giving birth in a private hospital was a barrier to using EBP. Women who gave birth in private hospitals were more likely not to use the EPB, such as: not having the compan-ionship of choice during childbirth (47.7%, p ≤ 0.05), doula support (69.9% p ≤ 0.01); midwife care (80.2%, p ≤ 0.01); freedom of mobility during labor (74.2%, p ≤ 0.01); choice of position during delivery (71.7%, p ≤ 0.01); and use of non-pharmacological methods for pain relief (90.0%, p ≤ 0.01) (Table 3).

The inadequate hospital ambiance is observed when the environment and infrastructure are not prepared to offer the use of EBP and was commonly described by women who gave birth in private hospitals, although a few women also encountered the same problem in public hospitals. Rigid hospital protocols or professional conduct, frequently not based on evidence, were also part of the overall inadequate hospital ambiance and described by women as a bar-rier to using the practices. Three women described similar experiences with barriers to using EBP while having had different types of birth and giving birth in different types of hospitals: 1) *"After my admission, I was informed that the delivery suite was closed down and I would*

*have to go through labor in a common room, with no access to non-pharmacological methods of pain relief."* (Cesarean; Private Hospital)*; 2) "I just wanted to give birth in the water, but the reference hospital is not prepared for it."* (Vaginal Birth; Public Hospital); and 3) *"I wanted to give birth in the bathtub, but the doctor said "você não é peixe[1]"to be in the water."* (Vaginal Birth; Private Hospital).

[1] **Cultural reference: The doctor disagreed with the woman giving birth in the water, using offensive language to persuade her. The doctor stated that the woman was not a fish to want to give birth in the water**.

Having private health insurance was correlated with not using non-pharmacological pain relief methods (91.7% p $\leq$ 0.01) (Table 4). Considering that the majority of women who joined the intervention had private health insurance, which is not reflective of the national data, the intervention unintentionally targeted the population most vulnerable to unnecessary cesarean procedures [43]. Having to pay out of pocket, adding cost to the childbirth on top of the already paid premiums and deductibles, were reported barriers only at private hospitals. The additional cost created barriers to accessing non-pharmacological pain relief methods, such as freedom of mobility during childbirth and choice of position at delivery. A woman describes in detail her lack of choice facing the cost barrier: *"I wanted to have a completely natural birth, give birth in the bathtub, but I was informed that I should pay a doctor to stay only with me if I wanted it that way. Since I did not have the financial conditions and already paid private health insurance, I chose to follow the health plan coverage."*

Women in the lower-income range (2 to < 5 MW) were less likely to use a birth plan (35.1%, p $\leq$ 0.05) and have midwife care (40.1%, p $\leq$ 0.01) compared to women with more than 10 MW (Table 4). Moreover, being a black woman was correlated with not using a birth plan (59.3%, p $\leq$ 0.01) and not having doula support (56.7%, p $\leq$ 0.01) (Table 4). Furthermore, women who did not have doula support and midwife care described a lack of choices and lack of individual support during childbirth as barriers. Women describe the absence of individual-ized care with doula support and/or midwife care in public and private hospitals: *1) "I just missed having a doula. There [the private maternity hospital] has doulas, but the problem was there was only one doula working that day. She was going back and forth between another woman who was giving birth at the same time and me. I think that [support] was lacking."* (Private Hospital); and *2) "When I needed to start pushing the baby, I was lying in the normal position because the hospital indicates that it has to be this way. There were no midwives to help me"* (Public Hospital).

The lack of continuous support was frequently mentioned as a barrier and related to a feel-ing of dissatisfaction. Since 2005, Brazil has a law that ensures women the uninterrupted sup-port of companionship of their choice during labor, delivery, and post-partum care while admitted to any hospital [44]. Women's discontent with the lack of continuous support is exemplified by the three reports, regardless of the type of birth or type of hospital: 1) *"When the pain started to get stronger, I was taken to a delivery room. I was left alone there for one hour. The pain was intense; I was afraid; I called for my mother, who was outside because she and my friend could not go in. It was my first birth, I looked around the room, and noth-ing I thought, imagined or witnessed about humanized birth was happening in that cold, large, lonely room"* (Vaginal Birth; Private Hospital); 2) *"The cesarean had no complications, but they did not let my sister-in-law in."* (Cesarean; Public Hospital); and 3) *"The only thing I did not like in the hospital was that I could not have a companion until my doctor arrived. My husband only came in to stay with me after my doctor arrived. It was over one hour with-out support. That was the traumatic part."* (Vaginal Birth; Public Hospital).

Lack of orientation by a health professional during prenatal care was typically described as a reason for not using an EBP, as exemplified by this woman who could not bond with her

**Table 4. Use of intrapartum EBP among women in the Senses of Birth intervention who completed the follow-up survey by age, education, income, race, and health insurance. Brazil, 2015–2016.**

| EBP | Age — Total N = 550² | | | Education — Total N = 552² | | | Income¹ — Total N = 519² | | | | | Race — Total N = 552² | | | Private Health Insurance — Total N = 553² | | |
|---|---|---|---|---|---|---|---|---|---|---|---|---|---|---|---|---|---|
| | 19-34 years N (%) | ≥35 years N (%) | P value | ≤12 years N (%) | >12 years N (%) | P Value | <2 MW N (%) | 2 to <5MW N (%) | 5 to <10MW N (%) | ≥10 MW N (%) | P value | White N (%) | Black N (%) | P value | Yes N (%) | No N (%) | P value |
| **Birth Plan** | | | | | | | | | | | | | | | | | |
| Yes | 252 (83.4) | 50 (16.6) | 0.608 | 51 (16.7) | 255 (83.3) | 0.000** | 43 (14.8) | 89 (28.6) | 83 (28.6) | 75 (25.9) | 0.002* | 156 (51.1) | 149 (48.9) | 0.014* | 257 (84.0) | 49 (16.0) | 0.001** |
| No | 202 (81.8) | 45 (18.2) | | 79 (32.2) | 166 (67.8) | | 58 (24.5) | 80 (35.1) | 52 (22.8) | 38 (16.7) | | 100 (40.7) | 146 (59.3) | | 179 (72.8) | 67 (27.2) | |
| Total | 454 (82.7) | 95 (17.3) | | 130 (23.7) | 421 (76.3) | | 101 (19.7) | 169 (32.6) | 135 26.0 | 113 (21.8) | | 256 (46.5) | 295 (53.5) | | 436 (79.0) | 116 (21.0) | |
| **Companionship during Childbirth** | | | | | | | | | | | | | | | | | |
| Yes | 383 (81.0) | 85 (19.0) | 0.075^ | 105 (23.2) | 347 (76.8) | 0.775 | 77 (18.2) | 130 (30.7) | 115 (27.2) | 101 (23.9) | 0.076^ | 212 47.0 | 239 (53.0) | 0.912** | 358 (79.4) | 93 (20.6) | 0.641 |
| No | 74 (89.2) | 9 (10.8) | | 20 (24.7) | 61 (75.3) | | 20 (25.3) | 30 (38.0) | 19 (24.1) | 10 (12.7) | | 38 (46.3) | 44 (53.7) | | 64 (77.1) | 19 (22.9) | |
| Total | 455 (82.7) | 93 (17.3) | | 125 (23.5) | 408 (76.5) | | 97 (19.3) | 160 (31.9) | 134 (26.7) | 111 (22.1) | | 250 (46.9) | 283 (53.1) | | 422 (79.0) | 112 (21.0) | |
| **Doula Support** | | | | | | | | | | | | | | | | | |
| Yes | 127 (87.0) | 19 (13.0) | 0.110 | 28 (19.2) | 118 (80.8) | 0.115 | 24 (17.6) | 41 (30.1) | 30 (22.1) | 41 (30.1) | 0.060^ | 80 (54.8) | 66 (45.2) | 0.017** | 115 (78.8) | 31 (21.2) | 0.983 |
| No | 318 (81.1) | 74 (18.9) | | 101 (25.7) | 292 (74.3) | | 75 (20.3) | 125 33.8 | 100 27.0 | 70 (18.9) | | 170 (43.3) | 223 (56.7) | | 310 (78.7) | 84 (21.3) | |
| Total | 445 (82.7) | 93 (17.3) | | 139 (23.9) | 410 (76.1) | | 99 (19.6) | 166 32.8 | 130 25.7 | 111 (21.9) | | 250 (46.4) | 289 (53.6) | | 425 (78.7) | 115 (21.3) | |
| **Midwife Care** | | | | | | | | | | | | | | | | | |
| Yes | 255 (87.6) | 36 (12.4) | 0.001** | 73 (24.8) | 221 (75.2) | 0.593 | 55 (19.9) | 111 (40.1) | 61 (22.0) | 50 (18.1) | 0.001** | 127 (43.3) | 166 (56.7) | 0.123* | 217 (73.8) | 77 (26.2) | 0.002* |
| No | 190 (76.9) | 57 (23.1) | | 56 (22.9) | 189 (77.1) | | 44 (19.2) | 55 (24.0) | 69 (30.1) | 61 (26.6) | | 123 (50.0) | 123 (50.0) | | 208 (84.6) | 38 (15.4) | |
| Total | 445 (82.7) | 93 (17.3) | | 129 (23.9) | 410 (76.1) | | 99 (19.6) | 166 (32.8) | 130 (25.7) | 111 (21.9) | | 250 (46.4) | 289 (53.6) | | 425 (78.7) | 115 (21.3) | |
| **Freedom of Mobility during labor** | | | | | | | | | | | | | | | | | |
| Yes | 205 (83.0) | 42 (17.0) | 0.814 | 39 (15.7) | 109 (84.3) | 0.002* | 36 (15.6) | 78 (33.8) | 64 (27.7) | 53 (22.9) | 0.881 | 123 (49.8) | 124 (50.2) | 0.358 | 195 (78.6) | 53 (21.4) | 0.343 |
| No | 147 (82.1) | 32 (17.9) | | 50 (27.8) | 130 (72.2) | | 32 (18.6) | 57 (33.1) | 46 (26.7) | 37 21.5 | | 82 (45.3) | 99 (54.7) | | 149 (82.3) | 32 (17.7) | |
| Total | 352 (82.6) | 74 (17.4) | | 89 (20.6) | 339 (79.2) | | 68 (16.9) | 135 (33.5) | 110 (27.3) | 90 (22.3) | | 205 (47.9) | 223 (52.1) | | 344 (80.2) | 85 (19.8) | |
| **Choice of Position at delivery** | | | | | | | | | | | | | | | | | |
| Yes | 206 84.1 | 39 (15.9) | 0.357 | 41 (16.7) | 205 (83.3) | 0.014* | 35 (15.2) | 77 (33.3) | 66 (28.6) | 53 (22.9) | 0.713 | 123 (50.2) | 122 (49.8) | 0.269 | 196 (79.7) | 50 (20.3) | 0.758 |
| No | 146 80.7 | 35 (19.3) | | 48 (26.4) | 134 (73.6) | | 33 (19.2) | 58 (33.7) | 44 (25.6) | 37 (21.5) | | 82 (44.8) | 101 (55.2) | | 148 (80.9) | 35 (19.1) | |
| Total | 352 82.6 | 74 (17.4) | | 89 (20.6) | 339 (79.2) | | 68 (16.9) | 135 (33.5) | 110 (27.3) | 90 (22.3) | | 205 (47.9) | 223 (52.1) | | 344 (80.2) | 85 (19.8) | |
| **Non-Pharmacological Pain Relief Methods** | | | | | | | | | | | | | | | | | |
| Yes | 265 (84.9) | 47 (15.1) | 0.025* | 64 (20.3) | 252 (79.7) | 0.817 | 54 (18.3) | 98 33.2 | 80 (27.1) | 63 (21.4) | 0.554 | 156 (49.5) | 159 (50.5) | 0.323 | 241 (76.3) | 75 (23.7) | 0.000** |
| No | 83 (75.5) | 27 (24.5) | | 23 (21.3) | 85 (78.7) | | 13 (12.5) | 35 (33.7) | 30 (28.8) | 26 (25.0) | | 48 (44.0) | 61 (56.0) | | 100 (91.7) | 9 (8.3) | |
| Total | 348 (82.5) | 74 (17.5) | | 87 (20.5) | 337 (79.5) | | 67 (16.8) | 133 (33.3) | 110 (27.6) | 89 (22.3) | | 204 (48.1) | 220 (51.9) | | 341 (80.2) | 84 (19.8) | |

¹Monthly Minimum Wage in 2015: R$788.00 = U$224.14.

²Total varies due to missing data for each variable.

Chi-Square

^P value ≤ 0.1

*P-value ≤ 0.05

**P-value ≤ 0.001.

prenatal care doctor due to the rotation of professionals: *"I did not even talk to my prenatal doctor, who was a family doctor about [the birth plan]. Each month I saw a different doctor. There was no time to bond with anyone."*

Women reported not writing a birth plan for different reasons: disbelief that it would be followed; a perception that it should only be used for normal birth; a lack of perceived self-efficacy; to avoid creating an expectation related to the birth; and lack of bond with the obstetrician. Some women reported having discussed the plan with their birth team or companions, but not creating a written document or in-depth plan, or not presenting it during childbirth at the hospital. Lack of time during pregnancy and lack of a recognized need for a birth plan were also frequently described as reasons for not writing one, as we can see in the examples below described by two women who had a cesarean at different types of hospitals: *"I just talked to the doctor about my expectations, but this was not documented in a birth plan. I think I felt embarrassed to make one."* (Private Hospital); *"Because I knew [the birth plan] was not going to be followed, [being in a] public hospital and all. Of course, I wanted to do it and follow it."* (Public Hospital).

### Facilitators and strategies to use the intrapartum EBP

Women reported that having their choices and desires respected was an important facilitator because it gave them the possibility to use their preferred practices. Women who believed they were able to have a normal birth after the SoB intervention were more likely to have doula support (97.9%, $p \leq 0.05$) and midwife care (95.9%, $p \leq 0.05$) (Table 3). Receiving individualized care was a strategy usually linked with the presence of doulas and midwives and was described as an incentive to use non-pharmacological methods for pain relief, freedom of mobility during labor, and choice of position at delivery: *"The midwife was available all the time. Very attentive. Then I did squats because she told me it would help to dilate."*. Having family support was a facilitator commonly described in conjunction with having the woman's desires and choices respected, as described by this woman referring to her husband's presence: *This moment is unforgettable and helped a lot to push my baby. I felt my little one be born. My [husband] was holding me inside the bathtub, helping me, gave me strength and support."*

Having the EBP offered and accessible was a critical facilitator described, often related to the adequate hospital ambiance, appropriate infrastructure, and individualized support offered, as exemplified by these two quotes: 1) *"It was beautiful!! My husband accompanied me all the time, and the facilities already had all the structure, so I did not have to use invasive methods to relieve the pain."*; and 2) *"The hospital has a labor and delivery room, where I had access to all the equipment for pain relief. My doula and my chosen companions were always massaging and encouraging."*

Having private health insurance was associated with the use of a birth plan (84.0%, $p \leq 0.01$) and midwife care (84.6%, $p \leq 0.01$) (Table 4). This likely represents the group of women who decided to pay out of pocket for a humanized birth team, as this woman reflects upon: *"So, [the birth plan was respected] only because I was with a private team, to tell you the truth. I had a doula that was with me the whole time."*

Discussing the birth plan with health professionals and trusting the health professionals were described as strategies/facilitators to have the birth plan respected. However, a few times, the trust and previous discussion with the obstetrician were reasons the women had no written birth plan, as one woman describes: *"She knows my choices and desires, I did not fill that form, but to me, it was like a birth plan."*

Age and education were also found to be facilitators to use the EBP when observing a bivariate analysis. Women who were younger than 34 years old were more likely to have midwife

care (87.6%, p ≤ 0.01) and use of non-pharmacological pain relief methods (84.9%, p ≤ 0.05) than women over 35 years (Table 4). Women who had more than 12 years of formal education were more likely to use a birth plan (83.3%, p ≤ 0.01), had freedom of mobility during childbirth (84.3%, p ≤ 0.05), and choose the position during delivery (83.3%, p ≤ 0.01) than women with less education (Table 4).

The SoB intervention was likely one of the facilitators to using the EBP as described by women. Three women who perceived an increase in EBP knowledge after the intervention described how the SoB intervention impacted their behavior: *1) "In fact, the Senses of Birth was part of my birth plan. All the information I got at the intervention was part of my entire pregnancy"*; 2) *"[The interventions influenced me], because the information was excellent. Especially about the doula."*; and 3) *It influenced me; I already knew that a companion's presence was important, and after participating in the Senses of Birth, this concept was more evident to me.*

### Women's perceived knowledge and the use of intrapartum EBP

An increase in knowledge was perceived by women who used the EBP, and women who did not use the EBP also presented an increase in their mean knowledge score after the intervention for all domains (Table 5). Nonetheless, women who did not use EBP had a lower mean knowledge score before and after exposure to the SoB intervention than those who used EBP (Table 5).

Women's knowledge change score (Fig 3) represents women's perception of knowledge variation after the intervention. The association of the change score with each of the intrapartum EBPs was observed among women who used and did not use the practices. There were no zero or negative change scores found, consistent with an overall positive impact of the intervention on women's knowledge for all three domains, regardless of whether or not they used each of the practices (Fig 3).

The EBP Knowledge total change score was higher than the change scores for Normal Birth Knowledge and Cesarean knowledge, indicating that women perceived a higher knowledge increase for EBP (Fig 3). Nevertheless, the change score among women who did not use the EBPs was higher than the change score for women who used the EBPs—indicating that acquiring knowledge at the SoB intervention was perhaps insufficient to overcome barriers to using the intrapartum EBPs.

## Discussion

In this study, the majority of women used intrapartum EBPs, except for receiving doula support. Using intrapartum EBPs was associated with high mean scores of perceived knowledge before the SoB intervention; giving birth in a public hospital, and having a vaginal birth. Giving birth in a private hospital was associated with not using EBP. Some practices were also associated with socioeconomic characteristics: women who had a higher income were more likely to use a birth plan and midwife care; being white was associated with the use of doula support; and having more than 13 years of formal education was associated with the use of a birth plan, freedom of mobility during labor, and freedom of choice of delivery position. Midwife care and doula support were associated with women who believed they were able to have a normal birth after participating in the SoB intervention during pregnancy.

Women who used the intrapartum EBP described a positive childbirth experience referring to feelings of satisfaction, safety, and respect. In contrast, negative childbirth experiences were referred to by women who did not use the practices with feelings of dissatisfaction, loneliness, and fear. Barriers to using intrapartum EBP identified by women were the absence of

**Table 5. Use of intrapartum evidence-based practices among women in the Senses of Birth intervention who participated in the follow-up survey by knowledge domain of EBP, normal birth, and cesarean mean score before and after the intervention.** Brazil, 2015–2016.

| | Normal Birth Knowledge | | | | | | Cesarean Knowledge | | | | | | EBP Knowledge | | | | | |
|---|---|---|---|---|---|---|---|---|---|---|---|---|---|---|---|---|---|---|
| | Mean Bef. SoB | SD | P-value | Mean Aft. SoB | SD | P value | Mean Bef. SoB | SD | P value | Mean Aft. SoB | SD | P value | Mean Bef. SoB | SD | P value | Mean Aft. SoB | SD | P Value |
| **Birth Plan[2, 3]** | | | | | | | | | | | | | | | | | | |
| Y | 4.02 | 0.85 | 0.000** | 4.66 | 0.53 | 0.000** | 3.89 | 0.92 | 0.000** | 4.55 | 0.60 | 0.000** | 3.60 | 1.05 | 0.000** | 4.42 | 0.66 | 0.000** |
| N | 3.64 | 0.92 | | 4.47 | 0.63 | | 3.41 | 0.98 | | 4.20 | 0.85 | | 2.94 | 1.02 | | 4.05 | 0.76 | |
| T | 3.85 | 0.90 | | 4.58 | 0.58 | | 3.68 | 0.97 | | 4.40 | 0.74 | | 3.30 | 1.09 | | 4.26 | 0.73 | |
| **Companionship during Childbirth[3]** | | | | | | | | | | | | | | | | | | |
| Y | 3.89 | 0.90 | 0.014* | 4.59 | 0.56 | 0.116^ | 3.74 | 0.95 | 0.003* | 4.43 | 0.71 | 0.004* | 3.38 | 1.08 | 0.002* | 4.30 | 0.71 | 0.002* |
| N | 3.63 | 0.93 | | 4.48 | 0.70 | | 3.39 | 1.06 | | 4.18 | 0.90 | | 2.97 | 1.08 | | 4.03 | 0.82 | |
| T | 3.85 | 0.91 | | 4.58 | 0.59 | | 3.69 | 0.98 | | 4.40 | 0.75 | | 3.31 | 1.09 | | 4.26 | 0.74 | |
| **Doula Support[3]** | | | | | | | | | | | | | | | | | | |
| Y | 4.09 | 0.81 | 0.000** | 4.65 | 0.51 | 0.052 | 3.90 | 0.97 | 0.001** | 4.48 | 0.77 | 0.093^ | 3.72 | 1.08 | 0.000** | 4.43 | 0.69 | 0.001** |
| N | 3.76 | 0.91 | | 4.54 | 0.61 | | 3.59 | 0.95 | | 4.36 | 0.73 | | 3.14 | 1.04 | | 4.19 | 0.74 | |
| T | 3.85 | 0.90 | | 4.57 | 0.59 | | 3.67 | 0.97 | | 4.39 | 0.74 | | 3.30 | 1.09 | | 4.25 | 0.74 | |
| **Midwife Care[3]** | | | | | | | | | | | | | | | | | | |
| Y | 3.91 | 0.87 | 0.071^ | 4.61 | 0.55 | 0.111^ | 3.75 | 0.93 | 0.061 | 4.44 | 0.69 | 0.115^ | 3.39 | 1.08 | 0.034* | 4.35 | 0.68 | 0.001** |
| N | 3.77 | 0.93 | | 4.53 | 0.62 | | 3.60 | 1.01 | | 4.34 | 0.80 | | 3.20 | 1.08 | | 4.14 | 0.79 | |
| T | 3.85 | 0.80 | | 4.57 | 0.59 | | 3.67 | 0.97 | | 4.39 | 0.74 | | 3.30 | 1.09 | | 4.25 | 0.74 | |
| **Freedom of Mobility during labor[2, 3]** | | | | | | | | | | | | | | | | | | |
| Y | 4.08 | 0.80 | 0.000** | 4.65 | 0.50 | 0.010* | 3.86 | 0.96 | 0.000** | 4.49 | 0.71 | 0.004* | 3.64 | 1.07 | 0.000** | 4.40 | 0.68 | 0.001** |
| N | 3.67 | 0.95 | | 4.50 | 0.67 | | 3.49 | 0.99 | | 4.28 | 0.84 | | 3.10 | 1.03 | | 4.17 | 0.74 | |
| T | 3.90 | 0.90 | | 4.59 | 0.58 | | 3.71 | 0.98 | | 4.40 | 0.77 | | 3.41 | 1.08 | | 4.30 | 0.71 | |
| **Choice of Position at delivery[2, 3]** | | | | | | | | | | | | | | | | | | |
| Y | 4.08 | 0.82 | 0.000** | 4.66 | 0.51 | 0.004* | 3.88 | 0.96 | 0.000** | 4.50 | 0.70 | 0.002* | 3.68 | 1.03 | 0.000** | 4.42 | 0.67 | 0.000** |
| N | 3.67 | 0.93 | | 4.50 | 0.66 | | 3.47 | 0.98 | | 4.27 | 0.84 | | 3.05 | 1.05 | | 4.15 | 0.74 | |
| T | 3.90 | 0.89 | | 4.59 | 0.58 | | 3.71 | 0.99 | | 4.40 | 0.77 | | 3.41 | 1.08 | | 4.30 | 0.71 | |
| **Non-Pharmacological Pain Relief Methods[3]** | | | | | | | | | | | | | | | | | | |
| Y | 3.99 | 0.87 | 0.000** | 4.60 | 0.56 | 0.400 | 3.80 | 0.98 | 0.001** | 4.42 | 0.77 | 0.590 | 3.56 | 1.07 | 0.000** | 4.37 | 0.67 | 0.001** |
| N | 3.63 | 0.90 | | 4.55 | 0.66 | | 3.44 | 0.99 | | 4.37 | 0.77 | | 3.01 | 1.02 | | 4.10 | 0.78 | |
| T | 3.90 | 0.89 | | 4.59 | 0.58 | | 3.71 | 0.99 | | 4.41 | 0.80 | | 3.42 | 1.09 | | 4.30 | 0.71 | |

[1]Mean scale before or after the SoB Intervention varies from 1 to 5 points.

[2]Welch Robust Test of Equality of Means–P-value ≤ 0.05.

[3]Levene Statistics—Test of Homogeneity of Variances–P-value ≥ 0.05.

[4]All variables meet assumption criteria of distribution of means. All variables presented a normal distribution.

Chi-Square

^P-value ≤ 0.1

*P-value ≤ 0.05

**P-value ≤ 0.001.

individualized support, non-respect for their choices, non-provision of EBP by professionals, lack of structure and ambiance in hospitals to use EBP, and rigid protocols not centered on women's needs and often not based on evidence. Facilitators and strategies reported by women focused on the increased perception of self-efficacy, perceived control over their bodies, having the practices offered, continuous individualized care, and having their choices and desires respected.

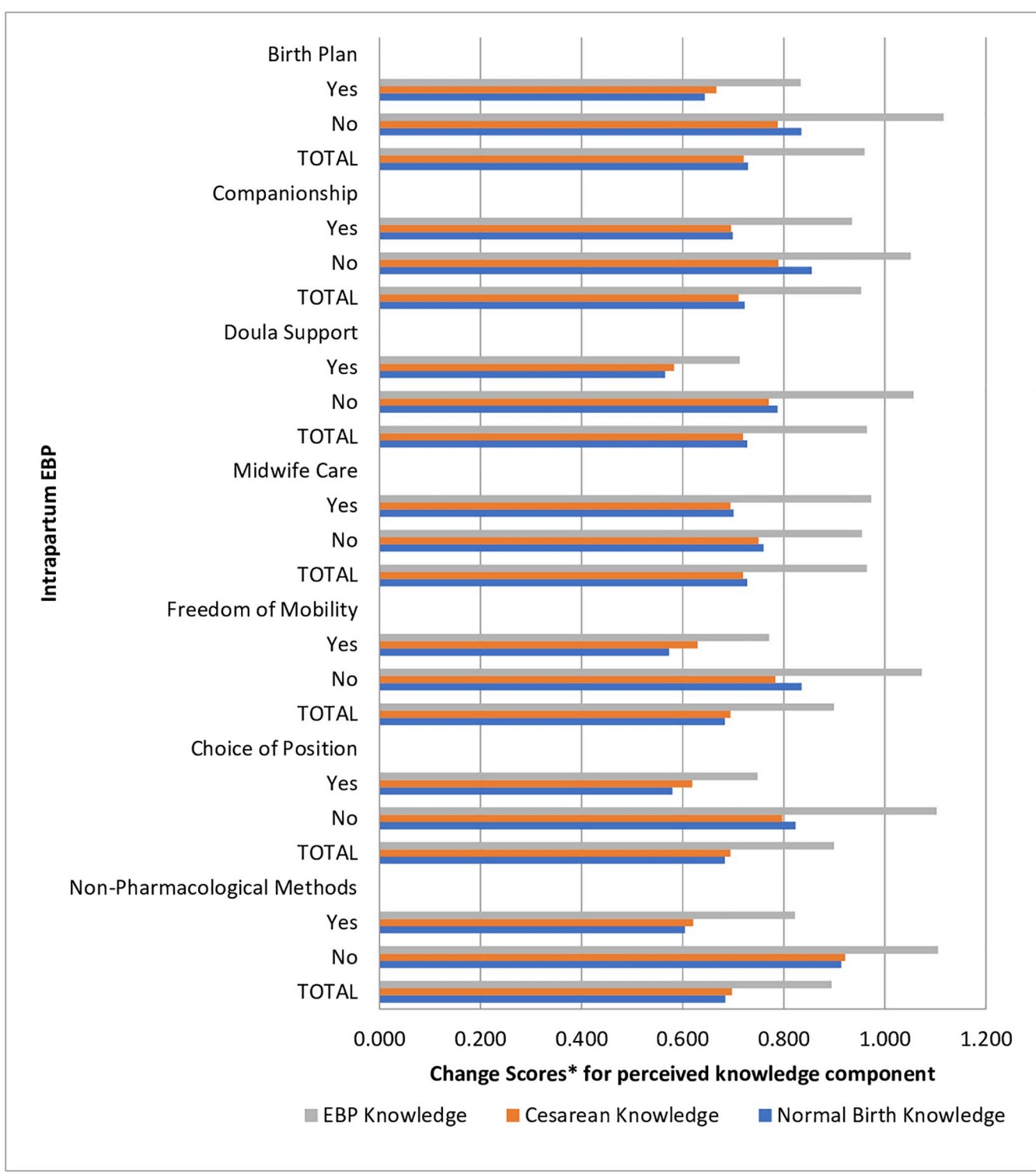

*Change score is the women's mean score for the perceived knowledge component after the SoB subtracted by the mean score before the SoB – possibly ranging from -5 to 5

**Fig 3. Women's change score* knowledge domain by use of intrapartum evidence-based practices after participating in the Senses of Birth intervention.** Brazil, 2015–2016.

## Using the intrapartum EBPs

Women in this study used the EBP more often than women nationwide as registered by the Birth in Brazil (BB) Study in 2014 (data collection in 2011–2012) [16, 45, 46], which could be partially attributed to the impact of the SoB intervention. Doula support was the intrapartum EBP with the lowest frequency of use among women who participated in the SoB intervention (26.9%). Nonetheless, the percentage of women with doula support in this study was higher than reported by women in the BB study, of which only 0.1% of women had doula support [45]. This noticeable increase in doula support during childbirth among Brazilian women who participated in the SoB study is also observed among midwife care. While 54.4% of women in the SoB study had midwife care, only 7.7% of women in the Birth Brazil Study had the same support [46]. The uninterrupted presence of a companion of choice was also higher among women who participated in the SoB intervention (81.6%) compared to the national BB study, 18.8% [45].

A recent nationwide study evaluated the Stork Network's implementation in public hospitals and the Healthy Birth program in private hospitals, with data collected in 2017, two years after this study about the SoB intervention [47]. Their results show that, among women who had a vaginal birth, 69.2% reported having the freedom of mobility during childbirth in public hospitals [47]. In SoB, 71.9% of women who had vaginal or cesarean births in public hospitals reported having the freedom of mobility.

Women who participated in the SoB intervention were more likely to use EBP when compared with the 2012 and 2017 studies, which may indicate an impact of the intervention supporting women to overcome barriers that the policies and programs in place have not yet achieved. The primary outcome of EBP use reported by women was satisfaction regardless of the type of birth. Satisfaction with the childbirth experience is one of the outcomes WHO describes when recommending a positive childbirth experience [3] and can contribute to a positive attitude towards using the EBPs.

As expected, having a vaginal birth was associated with the use of all the intrapartum EBP, which also allowed women who had a vaginal birth more opportunities to identify the barriers, facilitators and describe strategies used. In Brazil, the United States, and Spain, higher rates of doula support, midwife care, and use of a birth plan were each associated with having a vaginal birth [6, 48–50]. In contrast, women who had an intrapartum cesarean identified a wide range of barriers for using EBP, such as no access to non-pharmacological pain relief methods, lack of individualized care, no companionship during childbirth, and choices disrespected.

The frequent descriptions of barriers by women in this study, more frequent than facilitators, may indicate that women knew better care could be provided and corroborate a perception of a woman's desire to reclaim a positive experience in birth. The findings are in agreement with other studies that interviewed women in Brazil, in which women described being able to recreate childbirth as a human experience and regain confidence over their body's ability to give birth when given new possibilities of experience where before there was only the precarious choice between unnecessary cesarean or violent vaginal childbirth [1, 51].

**Women's knowledge, self-efficacy, and the use of intrapartum EBP.** The group of women who participated in this study can be identified as a group of well-informed women for the EBP, Normal Birth, and Cesarean knowledge domains. The women also showed improved awareness of their reproductive rights and choices after participating in the SoB intervention [41]. Access to quality information during prenatal care increased women's self-

efficacy and perceived control [37, 52–54], allowing women to question unnecessary interventions, move away from exclusively medical-centered care, and avoid the repetition of a TMTS model.

Self-efficacy and having one's choices respected were two robust strategies/facilitators to using EBP described by women in this study, giving them the chance of becoming protagonists of their childbirth. These strategies/facilitators have also been described by women engaged with the Brazilian childbirth humanization movement to overcome barriers to positive childbirth experience [51]. Intervention studies based on the TPB with pregnant women identified that women with a higher perceived self-efficacy reported higher awareness of their health status and well-being during pregnancy and advocated for their rights, choices, and needs more frequently [52, 55]. Increased self-efficacy was associated with lower use of analgesia intrapartum and having doula support in different studies [55, 56].

An increase in knowledge about normal birth, cesarean, and EBP is associated with a positive impact on women's use of EBP and were also described by women as strategies to use intrapartum EBP. Women's knowledge about childbirth risks is a predictor for choosing to try a vaginal birth after cesarean [57]. Knowledge can give women the chance to understand the childbirth physiology, overcome fears, and deconstruct myths around it [51]. Women who participated in the SoB intervention reported feeling safe and having comforting/calming experiences when using the intrapartum EBPs.

However, an increase in knowledge is not enough for using intrapartum EBP. Before the intervention, women with lower mean knowledge scores presented high change knowledge scores but did not use the EBP. Therefore, although the intervention may have had a more significant impact on them, they might not have achieved a threshold of knowledge sufficient to impact the behavior or overcome the identified barriers. A short period between visiting the SoB intervention and changing previous arrangements for the childbirth could be one of the reasons for this finding. Previous arrangements for the childbirth include systemic changes such as choosing a hospital that follows EBP protocols and/or has a normal childbirth ambiance, changing physicians for a birth team that supports normal childbirth, hiring a doula, and creating/using a birth plan.

Considering that 49.1% of the women who completed the follow-up survey were in their third trimester when visiting the SoB intervention, it is likely that women with low knowledge might benefit from additional education to recognize the benefits of the practices and how to access the practices. They may also need to receive information earlier in their pregnancy, to achieve greater opportunities and time to increase their perceived self-efficacy. Nonetheless, the systemic barriers many women face to use the EBP cannot be minimized and should be seen as a significant factor.

## A systemic view of barriers to using EBP

The majority of barriers described by women reflected institutional, professional, or health system barriers, corroborating findings of studies that used other stakeholders and health record data as sources [8–12, 58]. The lack of hospital ambiance for using EBP and the use of rigid hospital protocols not based on evidence are described as barriers by women in this study. They frequently refer to not having been offered a practice, such as non-pharmacological pain relief, and describe having their choices disrespected with a justification that the hospital protocol/rule did not support moving during labor or choosing a position during delivery, among other choices. Hospital protocols or health professionals' recommendations against the companion present during delivery were reasons described for not having the companionship of their choice with them at all times, that is, having their rights disrespected, which was also observed by 81.4% of the women in a different Brazilian study [13].

Private hospitals were associated with failing to use intrapartum EBP, which is worrisome since 54% of childbirths in Brazil occur at private hospitals, with an average of 83% of births being cesarean [24, 25, 59]. Additionally, financial costs to use EBPs was a barrier described only by women who gave birth in private hospitals. Few studies discuss the use of EBP in Brazil and the type of hospital. However, different authors have discussed the increased rates of cesarean in the private sector, creating a Brazilian childbirth paradox of care, in which low-risk women are exposed to TMTS scenarios [60–62]. Therefore, the low use of EBP at private hospitals corroborates the high rates of cesarean, indicating that women may have fewer opportunities to have a positive childbirth experience in those settings.

The differences between private and public hospitals described by women when using the EBP might also reflect the non-compliance of private hospitals with the national maternal and child health care policies, as observed by other studies [30, 43, 63]. Although there is a national agency that oversees private hospitals and health insurance, the agency's reach for enforcing national policies and guidelines in private hospitals is lower when compared to the influence those policies have over the public hospital system. Private hospitals are less likely to adhere to programs and policies that promote the adoption of evidence-based clinical protocols pertaining to childbirth [47].

Another barrier identified for the use of EBP was the disbelief among women that a birth plan could be respected in a public hospital, indicating that women perceive a lack of autonomy and choice to access care in a public hospital. Not having a written document for the birth plan minimizes the importance of a birth plan as a communication platform to ensure informed consent and protect women's reproductive rights. In Brazil, the birth plan has been used as an intervention or strategy to increase women's knowledge and self-efficacy, ensuring women could receive qualified information and open a channel of communication with the prenatal care physician [51, 64]. It is also an important instrument to generate interest among women to use other EBP, incentivize critical reflection about obstetric care, and engage the family/companion of choice [65].

Eighty-four percent of women who had a birth plan had private health insurance. On the other hand, having private health insurance was associated with no midwife support and no use of non-pharmacological pain relief methods during childbirth, which could be explained by the overmedicalization of childbirth in Brazil, alongside a physician-centered model of care [62, 66]. Although much has been done to improve outcomes for mothers and babies, the increasing medicalization of the maternal care system widens the gap between high and low resource settings and distances the interventions from women-centered care [3].

Overall, women have described the need for more information regarding EBP. Lack of knowledge among pregnant women should be approached as a systemic barrier since it was described in conjunction with no time for orientation and conversations during prenatal care in the private sector and lack of bond with the obstetrician considering the high rotation of physicians at the primary care level for the public sector. It is known that women who receive incomplete information regarding childbirth are less likely to express their preferences and are more likely to be subjected to severe pain and describe stress during childbirth [67]. Well-informed women could not only present their preferences and choices but also advocate for their reproductive rights, increase their autonomy and self-efficacy, and defend themselves from obstetric violence and discrimination [51, 68, 69].

Women in this study have systematically described institutional barriers to using the intrapartum EBP, reinforcing that the target behavior (the use of the practices) is not entirely under their control. Nevertheless, lack of knowledge, need for more information, disbelief, and perceived low-self-efficacy were also barriers reported by the same women, indicating that their intention to perform a behavior is impacted by their perceived behavior control. Perceived

behavior control can be described as the women's perceived self-efficacy or confidence in their ability to perform the behavior [34]. Such perception is built by internal factors (knowledge acquired and skills learned) and external factors (practical resources available, opportunities to use it, and the presence of other supportive conditions) [34].

## Social inequalities and the use of intrapartum EBP

Social inequalities were observed regarding race, income, and education, corroborating the literature that indicates that Brazilian women have different access to maternal care impacted by social, racial, and income inequalities [70]. According to the Brazilian Census Data, Brazilian women of childbearing age are more often black, with less than eight years of education, an average income of less than 2 MW, and use the public health system [71, 72]. Thus, our study sample differs from the population in that it represents a group with more years of formal education, higher family income, and higher use or access to private health insurance. It is possible that our sample over-represents women who are looking for a positive childbirth experience. Nonetheless, Brazilian women who participated in the SoB intervention and had lower income were more likely not to use the birth plan or have midwife care during childbirth. More years of formal education in the SoB study was associated with using a birth plan, freedom of mobility during labor, and freedom of choice of delivery position. Black women were less likely to use a birth plan and have doula support.

The National Gender Statistics report also reflected that black women are facing higher access barriers. This group has less access to prenatal care and appointments than white women, and significant racial inequality even though the country has high coverage of prenatal care [73]. The BB Study found that low-risk black women were less likely to have freedom of mobility during labor and use non-pharmacological pain relief methods when compared to white women [19], while over 12 years of formal education and being white were facilitators to having the companionship of choice during childbirth, use of non-pharmacological pain relief methods, and freedom of mobility during labor [19, 45].

Improvements to decrease inequalities through social programs and public policies have impacted the country's scenario in recent decades, improving conditions that directly affect women's and children's health implemented in the past decade [30]. Nonetheless, structural changes are needed to ensure that low-income women with fewer years of formal education and who are black have the same access to EBP and opportunities to have a positive childbirth experience.

Studies have shown that prenatal care education has a positive effect on women's confidence and ability to handle the birth process, while also diminishing fear, decreasing anxiety, increasing self-efficacy, and leading to higher perceived control during labor [37, 52–54]. On the other hand, previous research in Brazil indicated that poor prenatal care education practices might reinforce a medical-centered model of care that does not value health education as a potential qualifying element of care [74, 75]. Health education can be one of the strategies to support women using EBP. However, it is not sufficient to reduce an inequality gap that is caused from of a lifetime of social inequalities combined with structural societal racism and structural barriers to accessing quality care [76–80].

## Strengths and limitations

Women who completed the follow-up survey presented a high perceived knowledge before and after the intervention compared with the group of women who only completed the post-intervention survey, indicating they were a group of women already a group of women already knowledgeable about maternal health care rights. Therefore, it is likely that Brazilian women

with lower perceived knowledge were underrepresented in our sample. Nonetheless, hearing the voices of well-informed Brazilian women is a strength of this study, as their perspectives are key to understanding the role that women-centered care can play in overcoming barriers and increasing strategies/facilitators for the use of EBP, and promoting, in the long-run, cultural change of the social norms that surround childbirth.

When describing their childbirth experience, women might be influenced by intrinsic social desirability to focus only on positive outcomes of birth. However, the anonymity of women's responses and engagement with the topic likely diminishes this influence over the results. Few studies have included a large sample of Brazilian women and allowed them to freely describe their experiences regarding intrapartum EBPs. The large sample was possible due to the online self-administered survey, although the instrument does not promote qualitative data collection as in-depth as small interview groups. There are also limitations inherent to a post-intervention cross-sectional design, where participants answered about before and after knowledge and childbirth experiences at a single point in time, without a non-exposed comparison group. As such, the results found do not permit causal inference. On the other hand, the mixed-method analysis used in this study promoted a rich exploration of themes, allowing the detailed description of the women's experiences and meanings for the use of EBP.

## Conclusion

This study indicates that Brazilian women have restricted access to intrapartum EBPs. Although current policies have improved the availability of those practices, there are still systemic barriers such as lack of protocols, absence of guidance, and inappropriate ambiance–infrastructure that makes it difficult for women to achieve a positive childbirth experience. Social inequality barriers indicate the need for tailored strategies to reach black, low income, and less formally educated women. Furthermore, the results indicate the need to address the private health system's commitment to decreasing unnecessary cesarean rates and improve women's childbirth experience.

Women who used the intrapartum EBP described a sense of control over their bodies and self-efficacy to advocate for their chosen practices. This bolsters the idea that promoting positive childbirth experiences can create new paths to exercise reproductive rights, childbirth, motherhood, sexuality, and positive perception of the female body's capacity for childbirth. Women saw the strategies to overcome the barrier to using EBP as a path to become the protagonist of their childbirth and regain a sense of lost autonomy provoked when the care is not centered on the patient.

Increasing women's knowledge is part of the path to promote a positive childbirth experience. However, it should be done while simultaneously working with institutions/hospitals and health professionals to overcome the barriers identified. This study gave the women who participated a chance to critically reflect upon Brazil's maternal health care scenario and advocate for their choices, desires, and rights. Hence, it is clear that health education is an essential element to increase the use of WHO and MS recommended practices. However, health education cannot guarantee access to EBP. Therefore, there is a need to combine health education with changes in the maternal health model of care, promoting evidence-based patient-centered care and adequate hospital ambiance to access the EBP.

## Supporting information

**S1 File. The Senses of Birth intervention and the theory of planned behavior.**
(DOCX)

**S2 File. The Senses of Birth post-intervention and follow-up surveys to pregnant women–English version.**
(DOCX)

**S3 File. The Senses of Birth post-intervention and follow-up surveys to pregnant women–Portuguese version.**
(DOCX)

**S4 File. Quantitative measures used in the "Brazilian women's use of evidence-based practices in childbirth after participating in the Senses of Birth intervention: A mixed-methods study".**
(DOCX)

## Acknowledgments

The authors would like to acknowledge the contribution of all members of the Senses of Birth research group and all professors and health professionals who conceptualized and implemented the Senses of Birth intervention. This manuscript is part of the requirements for the degree completion of Doctor of Public Health at the University at Albany by the first author, a grantee of the Science without Borders from the National Council of Technological and Scientific Development (CNPq).

## Author Contributions

**Conceptualization:** Luísa da Matta Machado Fernandes, Sônia Lansky, Benjamin A. Shaw.

**Data curation:** Luísa da Matta Machado Fernandes, Hozana Reis Passos.

**Formal analysis:** Luísa da Matta Machado Fernandes.

**Funding acquisition:** Sônia Lansky.

**Investigation:** Sônia Lansky.

**Methodology:** Luísa da Matta Machado Fernandes.

**Supervision:** Sônia Lansky, Christine T. Bozlak, Benjamin A. Shaw.

**Validation:** Hozana Reis Passos, Christine T. Bozlak, Benjamin A. Shaw.

**Writing – original draft:** Luísa da Matta Machado Fernandes.

**Writing – review & editing:** Luísa da Matta Machado Fernandes, Sônia Lansky, Hozana Reis Passos, Christine T. Bozlak, Benjamin A. Shaw.

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
