## [Decision Letter · Decision Letter 0]

19 Nov 2020

PONE-D-20-15544

Brazilian women’s use of evidence-based practices in childbirth after participating in the Senses of Birth intervention: a mixed-methods study.

PLOS ONE

Dear Dr. da Matta Machado Fernandes,

Thank you for submitting your manuscript to PLOS ONE. After careful consideration, we feel that it has merit but does not fully meet PLOS ONE’s publication criteria as it currently stands. Therefore, we invite you to submit a revised version of the manuscript that addresses the points raised during the review process.

The manuscript has a lot of value to the readership of the journal and uses mixed-methods in an adequate way. However, the manuscript needs to be improved in several sections throughout the paper as noted by reviewer 1. If the authors do not agree with the reviewers' comments, they must provide a justification for their approach. Please address the requested changes and resubmit as soon as possible.

We look forward to receiving your revised manuscript.

Kind regards,

Abraham Salinas-Miranda, MD, PhD

Academic Editor

PLOS ONE

Journal Requirements:

3. Please clarify in your Ethics statement and the Methods section whether the IRBs specifically approved this study. Please also clarify how eligible participants were identified and recruited.

4. Please provide the survey questions also in the original language.

7. Please include a copy of Table 7 which you refer to in your text on page 13.

Additional Editor Comments:

Dear authors, the manuscript has a lot of value to the readership of the journal and uses mixed-methods in an adequate way. However, the manuscript needs to be improved in several sections throughout the paper as noted by reviewer 1. If the authors do not agree with the reviewers' comments, they must provide a justification for their approach. Please address the requested changes resubmit as soon as possible.

Reviewers' comments:

Reviewer's Responses to Questions

**Comments to the Author**

1. Is the manuscript technically sound, and do the data support the conclusions?

Reviewer #1: Yes

Reviewer #2: Yes

2. Has the statistical analysis been performed appropriately and rigorously? 

Reviewer #1: Yes

Reviewer #2: Yes

3. Have the authors made all data underlying the findings in their manuscript fully available?

Reviewer #1: No

Reviewer #2: No

4. Is the manuscript presented in an intelligible fashion and written in standard English?

Reviewer #1: Yes

Reviewer #2: Yes

5. Review Comments to the Author

Reviewer #1: Thank you for the opportunity to review the paper entitled, “Brazilian women’s use of evidence-based practices in childbirth after participating in the Sense of Birth intervention: a mixed-methods study.” The authors are to be commended for a mixed methods approach that focuses on understanding women’s self-efficacy and use of patient driven essential birth practices. The paper provides extensive analysis and provides interesting information. As this is a post-intervention assessment, without a comparison group, the strengths and limitations of this study design need to be further addressed. Additionally, there is an opportunity to streamline the findings; I have made suggestions where I think sections can be simplified.

Major:

1. In the framing of the paper, the authors use the phrase “evidence-based practices (EBP)” in referring to birth plans, companionship, midwife care, freedom of mobility during labor, choice of position during labor, doula support, and non-pharmacological pain relief methods. As acknowledged in their introduction, EBP normally referred to provider based practices (Lines 78-86). As readership may be accustomed to the provider point of view, I would strongly encourage the authors to ensure consistent framing as patient-enacted or patient-engaged evidence-based practices. Further, many of these practices align with the larger movement around Respectful Maternity Care (RMC) and “what women want campaign.” I think inclusion of these references in the introduction may help contextualize the Senses of Birth intervention and the experience of laboring women reported here.

2. Can the authors reflect how many of these specific patient-focused EBP are in control of the woman/family? How much of the EBP (i.e. midwife support, doula support or position of delivery) are solely up to the patient or require support from clinic policy, facility staff, etc.? A theory of change/directed acyclic graph would be helpful to think about all the individual characteristics, their connection to self-efficacy and then adherence to patient-centered evidence-based practices.

3. The study design is a post-intervention assessment and designed to look at characteristics and perceived knowledge and behavior. The strengths and limitations of this study design need to be further described.

4. In light of this study design, the findings must be interpreted cautiously. In lines 534-541, the authors state that, “the SoB intervention presented itself as one of the facilitators to use EBP described by women.” Even in the limitations section, the phrasing in line 765-767 is suggestive of causal language. I do not believe causation (attribution) can be given to the Senses of Birth study given this study design. Please revise this language.

5. Further, the sampling frame and approach to be further described. It is unclear how the 1,287 women who completed the post-test survey after the intervention were selected. Can the authors provide information on the number of people who participated in the intervention, were approached for the post-test survey, and what was the response rate? How are the 555 similar to or different to women who did not take the online survey? Finally, were the 258 women who were in the qualitative interviews a part of the 555 or the 1287? How were the 258 women selected for the qualitative interviews? A flow diagram would be helpful.

6. With respect to representation, in the introduction, the authors note that 26% of Brazilians have private insurance (line 104); however, in this population, in the study, 78% of the population have private insurance. This is just one example where a discussion of representation of the included population would be helpful.

7. The results are extensive and provide very interesting findings. While I appreciate the quotes and the mixed-methods approach, I believe the main messages of the findings are getting lost in the very details descriptions. I would encourage authors to use figures or quote boxes to streamline findings.

8. Some of the key findings focus on change in self-efficacy. If this is a change in self-efficacy score, can the authors specific the time frame of change? I think this change represents self-efficacy after the woman received the intervention and further out in time after receiving the intervention. If it was meant to be pre-intervention and post-intervention, the authors will need to explain how they can calculate this change as pre-intervention information is unknown.

Minor:

• The introduction includes a lot of background information on setting and location. I think this information would be more

relevant and streamlined in the Methods section.

• The detailed description of the Senses of Birth background (Lines 160-179) in the Education Intervention section can be referenced or streamlined to previously published methods papers on the Senses of Birth intervention.

• The details of every variable assessed in the study would be more helpful in an appendix (Lines 207-258). Consider streamlining this section.

• The Figures are illegible in my reviewer’s copy so I have not been able to provide feedback on them.

• Table 1 notes under the knowledge domains there is “increased perceived, no increased perceived.” This is unclear to readers.

• The data are available upon request; could the quantitative data be de-identified/anonymized and shared publicly? I understand qualitative data may be more difficult to anonymize.

Reviewer #2: Commentary

From the methodological point of view, it is a study with a mixed design of quanti-quali character. The sample is considered appropriate for both methods and the statistical analysis is adequate. PAHO considers that the average number of Caesarean sections acceptable should be 15%. However, we know that there are absolute and relative indications for a Cesarean section. Within these conditions, we know that some social determinants are important and could explain some percentage of these cesarean sections, such as poverty, access to food and poor prenatal care, among other factors. In light of these factors, it should be considered whether this indicator of 15% should be raised and not force a reduction in the number of cesarean sections. This implies emphasizing the social determinants and those direct causes that could influence the decision of the mother or the doctor to have a Cesarean section instead of a natural birth. Pregnant women should be intervened from their first birth control, trying to guide them towards a natural birth.

6. PLOS authors have the option to publish the peer review history of their article (what does this mean?). If published, this will include your full peer review and any attached files.

Reviewer #1: No

Reviewer #2: No

---

## [Author Response · Author response to Decision Letter 0]

26 Jan 2021

Responses to Editor’s requests:

The manuscript was reviewed to meet PLOS ONE's style requirements, including those for file naming.

2. We suggest you thoroughly copyedit your manuscript for language usage, spelling, and grammar.

The manuscript was thoroughly copyedited for language usage, spelling, and grammar by the manuscript co-authors who are native English speakers. 

3. Please clarify in your Ethics statement and the Methods section whether the IRBs specifically approved this study. Please also clarify how eligible participants were identified and recruited.

The Ethics statement (page 44) and the Material and Methods section (page 6) were edited to clarify that the current study was approved by a Brazilian IRB, from the Federal University of Minas Gerais, and by the U.S. IRB at the University at Albany. All women provided written informed consent, and eligible participants were identified at the entrance of the Senses of Birth Intervention when answering affirmatively in response to the current pregnancy question. They were then invited to join the study by one of the trained interviewers. The data and sample subsections have been edited to include the requested information (page 8).

4. Please provide the survey questions also in the original language.

The survey questions from this manuscript are available in Portuguese (original language) and in English (author’s translation), respectively, in Supplementary files 2 and 3.

5. We note that you have indicated that data from this study are available upon request. PLOS only allows data to be available upon request if there are legal or ethical restrictions on sharing data publicly.

We have reviewed the PLOS ONE requirement for data availability and uploaded the minimal anonymized data set necessary to replicate our quantitative study findings to the Dryad public repository (https://datadryad.org/stash); the current DOI of the data set is doi:10.5061/dryad.r7sqv9sb8. The data available correspond to the quantitative data used, however the qualitative data used can not be made available to the public, considering the sensitivity of the data. The majority of the women answering the open-ended questions that composed the qualitative data used their proper names and last names of family members, physicians, midwifes and doulas. Therefore, following the requirements of our IRB approval and our ethical commitment with the women who answered the survey, we cannot make the qualitative data publicly available without compromising the women´s and health professionals’ identities. In this case, de-identifying the data by taking the names out of the answers would compromise the answers and quality of the data analysis. Therefore, the qualitative data will be available upon request to the Senses of Birth Steering Committee who will ensure the open-ended questions are used according to the IRB requirements. The request to the steering committee can be directed to the co-principal investigators: Dr. Bernardo Oliveira, at the School of Education from the Federal University of Minas Gerais. Contact information: posfaebernardooliveira@gmail.com; and Dr. Sônia Lansky, at the Belo Horizonte´s Department of Health. Contact information: sonialansky@gmail.com.

6. Your ethics statement should only appear in the Methods section of your manuscript. If your ethics statement is written in any section besides the Methods, please delete it from any other section

The manuscript was reviewed, and the ethics statement only appears in the Methods section.

7. Please include a copy of Table 7 which you refer to in your text on page 13.

The Table numbering was a typo. The correct table referred to in the paragraph is Table 1, and the text was corrected to reflect it. 

Responses to Reviewer #1 Major Comments:

1. In the framing of the paper, the authors use the phrase “evidence-based practices (EBP)” in referring to birth plans, companionship, midwife care, freedom of mobility during labor, choice of position during labor, doula support, and non-pharmacological pain relief methods. As acknowledged in their introduction, EBP normally referred to provider-based practices (Lines 78-86). As readership may be accustomed to the provider point of view, I would strongly encourage the authors to ensure consistent framing as patient-enacted or patient-engaged evidence-based practices. Further, many of these practices align with the larger movement around Respectful Maternity Care (RMC) and “what women want campaign.” I think inclusion of these references in the introduction may help contextualize the Senses of Birth intervention and the experience of laboring women reported here.

The evidence-based practices discussed on this manuscript are defined by WHO and were recently summarized as part of the Intrapartum Care recommendations for a positive childbirth (2018). A reference was added to the subsection ‘The SoB Health Education Intervention’ (page 5 and 7) to clarify the origins of these EBPs. The manuscript discusses the practices from the women´s/patients’ perspectives, however, the practices are not exclusively patient-engaged or patient-enacted. Results related to knowledge increase were discussed in a previous publication, published in the BMC Pregnancy and Childbirth Journal and cited in the current manuscript (reference 41). The connection of women´s perspectives related to the use of the practices and the Theory of Planned Behavior was strengthened in the manuscript as part of the discussion section (page 39).

The Respectful Maternity Care (RMC) Movement is known as the Childbirth Humanization Movement in Brazil. The Senses of Birth exhibit discusses the maternal care scenario under the Childbirth Humanization Movement and respectful Maternity Care movement principles. The reference to the RMC movement was added to the paper and its connection with the SoB clarified for the reader (Introduction - page 5 and Material and Methods page 6).

2. Can the authors reflect how many of these specific patient focused EBP are in control of the woman/family? How much of the EBP (i.e. midwife support, doula support or position of delivery) are solely up to the patient or require support from clinic policy, facility staff, etc.? A theory of change/directed acyclic graph would be helpful to think about all the individual characteristics, their connection to self-efficacy and then adherence to patient-centered evidence-based practices.

All the EBP discussed in this manuscript are not fully under a woman´s control and the structural barriers identified by women reflect this point. A statement acknowledging the Theory of Planned Behavior construct of “perceived control” and identifying the lack of infrastructure as one of the external barriers that can impact perceived control and therefore the intention to perform the behavior was added to the manuscript (page 39). The theoretical framework of Senses of Birth Intervention and its impact on birth outcomes reflecting its connection with the TPB and the use of EBP was added to the Supplementary file 1, referred to on page 7. 

3. The study design is a post-intervention assessment and designed to look at characteristics and perceived knowledge and behavior. The strengths and limitations of this study design need to be further described.

The Strengths and Limitations section (pages 41 and 42) was expanded upon to further describe the limitations of the study design.

4. In light of this study design, the findings must be interpreted cautiously. In lines 534-541, the authors state that, “the SoB intervention presented itself as one of the facilitators to use EBP described by women.” Even in the limitations section, the phrasing in line 765-767 is suggestive of causal language. I do not believe causation (attribution) can be given to the Senses of Birth study given this study design. Please revise this language.

The manuscript language was revised to avoid attributing cause relation between the intervention and the use of EBP, considering the limitations of the study design. 

5. Further, the sampling frame and approach (need) to be further described. It is unclear how the 1,287 women who completed the post-test survey after the intervention were selected. Can the authors provide information on the number of people who participated in the intervention, were approached for the post-test survey, and what was the response rate? How are the 555 similar to or different to women who did not take the online survey? Finally, were the 258 women who were in the qualitative interviews a part of the 555 or the 1287? How were the 258 women selected for the qualitative interviews? A flow diagram would be helpful.

The data and sample subsection of the Methods section was added to clarify the sampling procedures (page 8), including a description of the selection of the 1287 women pregnant women that visited the SoB and the subset of 258 women who were included in the qualitative analysis. A new figure was added to the manuscript representing the data collection and sampling methods flow (figure 1, page 8). Further detailed information can be found in a previous publication that describes the full SoB intervention (reference number 32). The results section presenting women’s characteristics was enhanced to include a discussion of the comparability of the women who answered the follow-up survey with the women who answered the post-test survey (page 14). Further information relating the differences and similarities are the subject of a previous publication cited in the manuscript (reference 42). 

6. With respect to representation, in the introduction, the authors note that 26% of Brazilians have private insurance (line 104); however, in this population, in the study, 78% of the population have private insurance. This is just one example where a discussion of representation of the included population would be helpful.

The discussion of the representation of the population was included in the manuscript. One statement regarding private health insurance rates was added in the subsection of “Perceived Barriers to using Intrapartum EBP” (page 22), and another paragraph discussing the income and schooling differences between our sample and the population was included in the subsection “Social inequalities and the use of intrapartum EBP” (page 40).

7. The results are extensive and provide very interesting findings. While I appreciate the quotes and the mixed-methods approach, I believe the main messages of the findings are getting lost in the very detailed descriptions. I would encourage authors to use figures or quote boxes to streamline findings.

The structure of the manuscript integrating the women´s quotes with the quantitative results presented was an intentional format to create an equivalence among quantitative and qualitative results. This type of mixed-method composition ensures that the priority population perspective is not lost. 

8. Some of the key findings focus on change in self-efficacy. If this is a change in self-efficacy score, can the authors specify the time frame of change? I think this change represents self-efficacy after the woman received the intervention and further out in time after receiving the intervention. If it was meant to be pre-intervention and post-intervention, the authors will need to explain how they can calculate this change as pre-intervention information is unknown.

Change in self-efficacy was not discussed in the study and there were no scores used to measure self-efficacy. Self-efficacy emerged as one of the forty-five characterizing codes identified in the open-code qualitative analysis and was grouped into the category of Facilitators. Self-efficacy was perceived by women as a strategy and as a facilitator to using the EBP. The word ‘perception’ or ‘perceived’ were included in the text when referring self-efficacy, in order to clarify to the readers that the discussion refers to women’s perceptions of self-efficacy and not any measured scales. 

Responses to Reviewer #1 Minor Comments:

1. The introduction includes a lot of background information on setting and location. I think this information would be more relevant and streamlined in the Methods section

The authors have reviewed the Introduction section and agreed to shorten the information regarding prenatal care and health education, moving it to the discussion section (page 41), while maintaining the background information related to the definition of EBP, conceptualization of the positive childbirth, introductions to EBP barriers and gaps in the literature related to studies regarding women’s perception of the barriers, and discussion of the literature related to EBP use and the organization of Brazil’s health system. This information was kept in the Introduction considering that it is key to understanding the framing of the study. 

2. The detailed description of the Senses of Birth background (Lines 160-179) in the Education Intervention section can be referenced or streamlined to previously published methods papers on the Senses of Birth intervention.

The description of the Senses of Birth Intervention was streamlined and the previous publications, in English and Portuguese, were referenced as sources for further information on the background of this intervention (page 6 – references 32 and 33).

3. The details of every variable assessed in the study would be more helpful in an appendix (Lines 207-258). Consider streamlining this section

The detailed information of all variables used and how they were prepared are now presented in supplementary file 4, referred to on page 10. The information regarding the groups of quantitative variables used was streamlined in the manuscript, while still maintaining the structure to ensure readers comprehension (pages 9-10). 

4. The Figures are illegible in my reviewer’s copy, so I have not been able to provide feedback on them.

The figures have been uploaded using new files, ensuring compliance with the PLOS ONE formatting requirements. 

5. Table 1 notes under the knowledge domains there is “increased perceived, no increased perceived.” This is unclear to readers.

The labels in Table 1 have been adjusted to “Perceived Knowledge Increased” and “Perceived Knowledge Not Increased” to improve reader’s understanding (page 14). 

6. The data are available upon request; could the quantitative data be de-identified/anonymized and shared publicly? I understand qualitative data may be more difficult to anonymize.

We have reviewed PLOS ONE requirements for data availability and uploaded the minimal anonymized data set necessary to replicate our quantitative study findings to the Dryad public repository (https://datadryad.org/stash) and the current DOI of the data set is doi:10.5061/dryad.r7sqv9sb8. The data available correspond to the quantitative data used, however the qualitative data used can not be made publicly available considering the sensitivity of the data. The majority of the women answering the open-ended questions that composed the qualitative data used proper names and last names of family members, physicians, midwifes and doulas. Therefore, following the requirements of our IRB approval and our ethical commitment with the women who answered the survey, we cannot make the qualitative data public available without compromising women´s and health professional’s identities. The qualitative data will be available upon request to the Senses of Birth Steering Committee, ensuring that the open-ended questions are used according to the IRB requirements.

---

## [Decision Letter · Decision Letter 1]

5 Mar 2021

Brazilian women’s use of evidence-based practices in childbirth after participating in the Senses of Birth intervention: a mixed-methods study.

PONE-D-20-15544R1

Dear Dr. da Matta Machado Fernandes,

We’re pleased to inform you that your manuscript has been judged scientifically suitable for publication and will be formally accepted for publication once it meets all outstanding technical requirements.

Kind regards,

Abraham Salinas-Miranda, MD, PhD

Academic Editor

PLOS ONE

Additional Editor Comments (optional):

The revisions were satisfactory to the reviewers and this paper is ready to be accepted for publication.

Reviewers' comments:

Reviewer's Responses to Questions

**Comments to the Author**

1. If the authors have adequately addressed your comments raised in a previous round of review and you feel that this manuscript is now acceptable for publication, you may indicate that here to bypass the “Comments to the Author” section, enter your conflict of interest statement in the “Confidential to Editor” section, and submit your "Accept" recommendation.

Reviewer #1: All comments have been addressed

Reviewer #2: All comments have been addressed

2. Is the manuscript technically sound, and do the data support the conclusions?

Reviewer #1: (No Response)

Reviewer #2: Yes

3. Has the statistical analysis been performed appropriately and rigorously? 

Reviewer #1: (No Response)

Reviewer #2: Yes

4. Have the authors made all data underlying the findings in their manuscript fully available?

Reviewer #1: (No Response)

Reviewer #2: Yes

5. Is the manuscript presented in an intelligible fashion and written in standard English?

Reviewer #1: (No Response)

Reviewer #2: Yes

6. Review Comments to the Author

Reviewer #1: (No Response)

Reviewer #2: Temas metodológicos fueron superados ampliando las explicaciones solicitadas y se agregaron las limitaciones observadas en el estudio. Esto ayudará a tener una mejor comprensión de los alcances del mismo estudio. Se aclaró el tema de la muestra y corrigieron algunos términos. Se aclaró los temas relacionados al comité de ética.

7. PLOS authors have the option to publish the peer review history of their article (what does this mean?). If published, this will include your full peer review and any attached files.

Reviewer #1: No

Reviewer #2: **Yes: **Jairo Vanegas López

---

## [Editor Report · Acceptance letter]

6 Apr 2021

PONE-D-20-15544R1 

Brazilian women’s use of evidence-based practices in childbirth after participating in the Senses of Birth intervention: a mixed-methods study. 

Dear Dr. da Matta Machado Fernandes:

I'm pleased to inform you that your manuscript has been deemed suitable for publication in PLOS ONE. Congratulations! Your manuscript is now with our production department. 

Kind regards, 

on behalf of

Dr. Abraham Salinas-Miranda 

Academic Editor

PLOS ONE